# Directed Cyclic Graph for Causal Discovery from Multivariate Functional Data

**Saptarshi Roy**
Department of Statistics
Texas A&M University
College Station, TX 77843
roys8001@stat.tamu.edu

**Raymond K. W. Wong**
Department of Statistics
Texas A&M University
College Station, TX 77843
raywong@tamu.edu

**Yang Ni**
Department of Statistics
Texas A&M University
College Station, TX 77843
yni@stat.tamu.edu

## Abstract

Discovering causal relationship using multivariate functional data has received a significant amount of attention very recently. In this article, we introduce a functional linear structural equation model for causal structure learning when the underlying graph involving the multivariate functions may have cycles. To enhance interpretability, our model involves a low-dimensional causal embedded space such that all the relevant causal information in the multivariate functional data is preserved in this lower-dimensional subspace. We prove that the proposed model is causally identifiable under standard assumptions that are often made in the causal discovery literature. To carry out inference of our model, we develop a fully Bayesian framework with suitable prior specifications and uncertainty quantification through posterior summaries. We illustrate the superior performance of our method over existing methods in terms of causal graph estimation through extensive simulation studies. We also demonstrate the proposed method using a brain EEG dataset.

## 1 Introduction

**Motivation.** Multivariate functional data arise in many fields such as biomedical research [Wei and Li, 2008, Chiou and Müller, 2016], environmental science [Korte-Stapff et al., 2022], finance [Kowal et al., 2017], plant science [Wong et al., 2019, Park et al., 2022], and sport science [Volkmann et al., 2021] where multiple variables are measured over time or other domains. The increasing availability of functional data in these fields provides us with great opportunities to discover causal relationships among random functions for the better understanding of complex systems, which is helpful for various machine learning and statistics tasks such as representation learning [Schölkopf et al., 2021], fairness [Tang et al., 2023], transfer learning [Rojas-Carulla et al., 2018], and reinforcement learning [Zeng et al., 2023]. One motivating example is electroencephalography (EEG) where electrical activity from the brain is recorded non-invasively from electrode channels by placing them on the scalp or directly on the surface of the brain. Given its continuous nature and the short time separation between the adjacent measuring points, it is natural to treat the data at each brain location/region as a function over time. A relevant scientific goal is to estimate brain effective connectivity among different regions, which will potentially allow us to make better decisions, design more effective interventions, and avoid unintended consequences. However, existing structural equation model (SEM) based causal discovery methods assume acyclic relationships among the random functions by imposing a directed acyclic graph (DAG) structure, which may be too restrictive for many real applications. For example, there are strong indications that in brain effective connectivity studies, due to reciprocal polysynaptic connections, the brain regions are far from exhibiting acyclicity [Friston, 2011, Markov et al., 2012], and that in genetic pathways, due to the presence of multiple upstream regulators and downstream targets for every signaling component, feedback loops/directed cycles are regular motifs [Brandman

and Meyer, 2008]. Thus, in light of the prevalence of cycles in complex systems, it is desirable to have a flexible model for causal discovery among random functions that can account for such cyclic causal structures.

**Challenges.** Causal discovery for multivariate functional data in the presence of cycles is an inherently difficult problem that is not yet well understood. We highlight three prominent challenges. (i) Functional data are infinite-dimensional in nature. It may so happen that the low-frequency spectrum of one curve might causally influence the high-frequency spectrum of another curve. This demands identification of pertinent features that can be used to create a finite-dimensional representation of the data, which is easier to work with and analyze. However, the challenge is that we may not know *a priori* what the relevant features are when dealing with infinite-dimensional objects. Blind adoption of standard (non-causal-adaptive) low-dimensional features can lead to errors or inaccuracies. (ii) Although the identifiability of causal models for multivariate functional data in the absence of cycles has been established in recent works [Zhou et al., 2022b, Lee and Li, 2022], showing identifiability of causal models from multivariate data, let alone multivariate functions, is still a challenging and complex task in cases where causal relationships are obscured by the presence of cycles. (iii) It is common that functional data are only observed over discrete time points with additional noises. Such incomplete and noisy observations of the functions add another layer of difficulty in probing the causal relationships of interest.

**Related work.** Causal discovery from multivariate functional data has been studied by a few recent works [Zhou et al., 2022b, Lee and Li, 2022, Yang and Suzuki, 2022], which have already shown some promising results in discovering causality in, e.g., EEG data and longitudinal medical record data. However, all of them are limited to DAGs, which do not allow inference of cyclic causality. While there has been a surge of research on causal discovery methods for scalar random variables in the presence of feedback loops/cycles over the last few decades [Richardson, 1996, Lacerda et al., 2008, Mooij et al., 2011, Hyttinen et al., 2012, Huang et al., 2019, Mooij and Heskes, 2013, Mooij and Claassen, 2020, Zhou et al., 2022a], none of these approaches have been extended to discovering causal dependencies among random functions in multivariate settings. Therefore, how to handle cyclic causal relationships among multivariate functional data while addressing the aforementioned challenges remains a largely unsolved problem.

**Contributions.** In this paper, we propose an operator-based non-recursive linear structural equation based novel causal discovery framework that identifies causal relationships among functional objects in the presence of cycles and additional measurement/sampling noises. Our major contribution is four-fold.

1. We consider a causal embedding of the functional nodes into a lower-dimensional space for dimension reduction that adapts to causal relationships.

2. We prove that the causal graph of the proposed model is uniquely identifiable under standard causal assumptions.

3. We capture within-function dependencies using a data-driven selection of orthonormal basis that is both interpretable and computationally efficient.

4. To perform inference and uncertainty quantification from finite-sample data, we adopt a fully Bayesian hierarchical formulation with carefully selected prior distributions. Posterior inference is performed using Markov chain Monte Carlo (MCMC). We demonstrate the effectiveness of the proposed method in identifying causal structure and key parameters through simulation studies and apply the framework to the analysis of brain EEG data, illustrating its real-world applicability. Code will be made available on the project's website on Github.

## 2  Model definition and causal identifiability

### 2.1  Notations

Let $[p] = \{1, \ldots, p\}$ for any positive integer $p$. A causal directed cyclic graph (DCG) is a graph $\mathcal{G} = (\mathcal{V}, \mathcal{E})$, which consists of a set of vertices or nodes $\mathcal{V} = [p]$ representing a set of random variables

and a set of directed edges $\mathcal{E} = \{\ell \to j | j, \ell \in \mathcal{V}\}$ representing the direct causal relationships among the random variables. In a DCG, we do not assume the graph to be acyclic. A causal DCG model is an ordered pair $(\mathcal{G}, \mathbb{P})$ where $\mathbb{P}$ is a joint probability distribution over $\mathcal{V}$ (more rigorously, the random variables that $\mathcal{V}$ represents) that satisfies conditional independence relationships encoded by the causal DCG $\mathcal{G}$. A simple directed cycle is a sequence of distinct vertices $\{v_1, \ldots, v_k\}$ such that the induced subgraph by these vertices is $v_1 \to \cdots \to v_k \to v_1$. For a vertex $j \in \mathcal{V}$, we use $\mathrm{pa}(j)$ to denote the set of parents (direct causes).

## 2.2 Model framework

Consider a multivariate stochastic process $\boldsymbol{Y} = (Y_1, \ldots, Y_p)^\top$ where each $Y_j$ is defined on a compact domain $\mathcal{T}_j \subset \mathbb{R}$. Without loss of generality, we assume $\mathcal{T}_1 = \cdots = \mathcal{T}_p = [0, 1]$. Suppose $Y_j \in \mathcal{H}_j$ where $\mathcal{H}_j$ is a Hilbert space of functions defined on $\mathcal{T}_j$. We let $\langle \cdot, \cdot \rangle$ denote the inner product of $\mathcal{H}_j$. We propose a causal model that captures the relationships among $Y_1, \ldots, Y_p$.

Our proposed model considers an operator-based non-recursive linear structural equation model on the random functions $\boldsymbol{Y}$ as

$$Y_j(\cdot) = \sum_{\ell \in \mathrm{pa}(j)} (\mathcal{B}_{j\ell} Y_\ell)(\cdot) + f_j(\cdot), \quad \forall j \in [p], \tag{1}$$

where $\mathcal{B}_{j\ell}$ is a linear operator that maps $\mathcal{H}_\ell$ to $\mathcal{H}_j$, and $f_j \in \mathcal{H}_j$ is an exogenous stochastic process. Clearly, for any $j, \ell \in \mathcal{V}$ such that the edge $\ell \to j \in \mathcal{E}$, $\mathcal{B}_{j\ell}$ is not a null operator. Now by stacking the $p$ equations in (1), we obtain

$$\boldsymbol{Y} = \mathfrak{B}\boldsymbol{Y} + \boldsymbol{f}, \tag{2}$$

where $\mathfrak{B} = (\mathcal{B}_{j\ell})_{j,\ell=1}^p$ is a matrix of operators and $\boldsymbol{f} = (f_1, \ldots, f_p)^\top$ is a $p$-variate stochastic process. In DAGs, the causal effect matrix can be arranged into a lower block triangular structure given a topological/causal ordering. But since our model allows for cycles, we have no such restriction on the structure of the operator matrix $\mathfrak{B}$ except that $\mathcal{B}_{jj}, \forall j \in [p]$, is null, i.e., no self-loops.

Model (1) is infinite-dimensional and hence challenging to estimate and interpret. To alleviate such difficulties, we consider a low-dimensional causal embedding structure. Specifically, we assume that the causal relationships are preserved in an unknown low-dimensional subspace $\mathcal{D}_j$ of $\mathcal{H}_j$. Denote the dimension of $\mathcal{D}_j$ by $K_j$. Let $\mathcal{P}_j$ and $\mathcal{Q}_j$ be the projection onto $\mathcal{D}_j$ and its orthogonal complement in $\mathcal{H}_j$ respectively. We assume $\mathcal{B}_{j\ell} = \mathcal{P}_j \mathcal{B}_{j\ell} \mathcal{P}_\ell$, which implies that causal effects can be fully described within the low-dimensional subspaces $\{\mathcal{D}_j\}_{j=1}^p$. As such, (1) can be split into

$$\mathcal{P}_j Y_j = \sum_{\ell \in \mathrm{pa}(j)} \mathcal{B}_{j\ell}(\mathcal{P}_\ell Y_\ell) + \mathcal{P}_j f_j, \tag{3}$$

$$\mathcal{Q}_j Y_j = \mathcal{Q}_j f_j.$$

We assume that $\mathcal{P}_j f_j$ and $\mathcal{Q}_j f_j$ are independent of each other. Now, by defining $\alpha_j = \mathcal{P}_j Y_j$ and $\epsilon_j = \mathcal{P}_j f_j, \forall j \in [p]$, (3) can be compactly written as

$$\boldsymbol{\alpha} = \mathcal{B}\boldsymbol{\alpha} + \boldsymbol{\epsilon}, \tag{4}$$

where $\boldsymbol{\alpha} = (\alpha_1, \ldots, \alpha_p)^\top$ and $\boldsymbol{\epsilon} = (\epsilon_1, \ldots, \epsilon_p)^\top$ with $\alpha_j, \epsilon_j \in \mathcal{D}_j, \forall j \in [p]$.

In practice, the random functions in $\boldsymbol{Y}$ can only be observed over a finite number of (input) locations, possibly with measurement errors. More specifically, for each random function $Y_j$, we observe $\{(t_{ju}, X_{ju})\}_{u=1}^{m_j}$, where $X_{ju} \in \mathbb{R}$ is the measurement of $Y_j$ at location $t_{ju} \in \mathcal{T}_j$ and $m_j$ is the number of measurements obtained from $Y_j$. Defining $\beta_j = \mathcal{Q}_j Y_j$, we consider the following measurement model:

$$\begin{aligned} X_{ju} &= Y_j(t_{ju}) + e_{ju} \\ &= \alpha_j(t_{ju}) + \beta_j(t_{ju}) + e_{ju}, \quad \forall u \in [m_j], j \in [p], \end{aligned} \tag{5}$$

with independent noises $e_{ju} \sim N(0, \sigma_j), \forall u \in [m_j]$.

More compactly, (5) can be written as

$$\boldsymbol{X} = \boldsymbol{\alpha}(\boldsymbol{t}) + \boldsymbol{\beta}(\boldsymbol{t}) + \boldsymbol{e}, \tag{6}$$

where $\boldsymbol{X} = (\boldsymbol{X}_1^\top, \ldots, \boldsymbol{X}_p^\top)^\top$, $\boldsymbol{\alpha}(\boldsymbol{t}) = (\boldsymbol{\alpha}_1(\boldsymbol{t}_1)^\top, \ldots, \boldsymbol{\alpha}_p(\boldsymbol{t}_p)^\top)^\top$, $\boldsymbol{\beta}(\boldsymbol{t}) = (\boldsymbol{\beta}_1(\boldsymbol{t}_1)^\top, \ldots, \boldsymbol{\beta}_p(\boldsymbol{t}_p)^\top)^\top$ and $\boldsymbol{e} = (\boldsymbol{e}_1^\top, \ldots, \boldsymbol{e}_p^\top)^\top$ with $\boldsymbol{X}_j = (X_{j1}, \ldots, X_{jm_j})^\top, \boldsymbol{\alpha}_j(\boldsymbol{t}_j) = (\alpha_j(t_{j1}), \ldots, \alpha_j(t_{jm_j}))^\top, \boldsymbol{\beta}_j(\boldsymbol{t}_j) = (\beta_j(t_{j1}), \ldots, \beta_j(t_{jm_j}))^\top$ and $\boldsymbol{e}_j = (e_{j1}, \ldots, e_{jm_j})^\top$.

We call our proposed model, **FENCE**, which stands for 'Functional Embedded Nodes for Cyclic causal Exploration', reflecting its purpose.

## 2.3 Causal identifiability

In this section, we shall show that the graph structure of the proposed FENCE model is identifiable for functional data measured discretely with random noises under several causal assumptions. We start by defining causal identifiability and state our assumptions.

**Definition 2.1.** (*Causal Identifiability*) Suppose $\boldsymbol{Y}$ is a $p$-variate random function and $\boldsymbol{X}$ is the observed noisy version of $\boldsymbol{Y}$ given by (6). Assume $\boldsymbol{X}$ follows FENCE model $\mathcal{S} = (\mathcal{G}, \mathbb{P})$ where $\mathcal{G}$ is the underlying graph and $\mathbb{P}$ is the joint distribution of $\boldsymbol{X}$ over $\mathcal{G}$. We say that $\mathcal{S}$ is causally identifiable from $\boldsymbol{X}$ if there does not exist any other $\mathcal{S}^* = (\mathcal{G}^*, \mathbb{P}^*)$ with $\mathcal{G}^* \neq \mathcal{G}$ such that the joint distribution $\mathbb{P}^*$ on $\boldsymbol{X}$ induced by $\mathcal{G}^*$ is equivalent to $\mathbb{P}$ induced by $\mathcal{G}$.

In other words, for a causal graph to be identifiable, there must not exist any other graph such that the joint distributions induced by the two different graphs are equivalent. Next, we list and discuss a few assumptions to establish the causal identifiability of the proposed model.

**Assumption 1.** (*Causal Sufficiency*) *The model $\mathcal{S} = (\mathcal{G}, \mathbb{P})$ is causally sufficient, i.e., there are no unmeasured confounders.*

Assuming no unmeasured confounders keeps the causal discovery task more manageable especially for cyclic graphs with purely observational data.

**Assumption 2.** (*Disjoint Cycles*) *The cycles in $\mathcal{G}$ are disjoint, i.e., no two cycles in the graph have two nodes that are common to both.*

Assuming disjoint cycles induces a natural topological ordering and forms a directed acyclic hypergraph-like structure within the DCG. The same assumption was made in Lacerda et al. [2008].

**Assumption 3.** (*Stability*) *For the model $\mathcal{S}$, the moduli of the eigenvalues of the finite rank operator $\mathcal{B}$ are less than or equal to $1$, and none of the real eigenvalues are equal to $1$.*

According to Fisher, 1970, the SEM in (4) can be viewed as being in a state of equilibrium, where the finite rank operator $\mathcal{B}$ represents coefficients in a set of dynamical equations that describe a deterministic dynamical system observed over small time intervals as the time lag approaches zero. The eigenvalue conditions are deemed necessary and sufficient for the limiting behavior to hold, as argued by Fisher, 1970. Such an assumption is widely adopted in e.g., econometrics, and Lacerda et al., 2008 made this assumption as well.

**Assumption 4.** (*Non-Gaussianity*) *The exogenous variables have independent mixture of Gaussian distributions. i.e., $\epsilon_{jk} \overset{ind}{\sim} \sum_{m=1}^{M_{jk}} \pi_{jkm} N(\mu_{jkm}, \tau_{jkm})$ with $M_{jk} \geq 2$.*

The assumption of non-Gaussianity on the exogenous variables has been proven useful in causal discovery as it induces model identifiability in the linear SEM framework [Aapo and Petteri, 1999, Shimizu et al., 2006, Lacerda et al., 2008, Spirtes and Zhang, 2016]. Mixture of Gaussian can approximate any continuous distribution arbitrarily well given a sufficiently large number of mixture components [Titterington et al., 1985, McLachlan and Peel, 2000, Rossi, 2014]. It is also easy to sample, which facilitates our posterior inference.

**Assumption 5.** (*Non-causal dependency*) *We assume $\boldsymbol{\beta}(\boldsymbol{t}) = \boldsymbol{C}(\boldsymbol{t})\boldsymbol{\gamma}$, where $\boldsymbol{C}(\boldsymbol{t}) = diag(\boldsymbol{C}_{11}(\boldsymbol{t}_1), \ldots, \boldsymbol{C}_{pp}(\boldsymbol{t}_p))$ and $\boldsymbol{\gamma}$ represent another exogenous component of the model. Here $\boldsymbol{C}_{jj}(\boldsymbol{t}_j)$ is a mixing matrix that mixes the independent entires in $\boldsymbol{\gamma}$ to generate temporal dependence within the $j$-th block. We assume $\gamma_{jk} \overset{ind}{\sim} \sum_{m=1}^{M_{jk}} \pi'_{jkm} N(\mu'_{jkm}, \tau'_{jkm})$ with $M_{jk} \geq 1$.*

Since the model assumes that all causal information in $\boldsymbol{Y}$ is preserved in the lower-dimensional space $\mathcal{D}_j$ and not in its orthogonal complement, it is apparent that while each $\boldsymbol{\beta}_j(\boldsymbol{t}_j)$ within a block can have temporal dependence, it is independent of $\boldsymbol{\beta}_\ell(\boldsymbol{t}_\ell)$ when $j \neq \ell$ and $j, \ell \in [p]$.

For some basis $\{\phi_{jk}\}_{k=1}^{K_j}$ that spans the low-dimensional causal embedded space $\mathcal{D}_j$, $\alpha_j$ in (5) can be further expanded by, $\alpha_j(t_{ju}) = \sum_{k=1}^{K_j} \tilde{\alpha}_{jk}\phi_{jk}(t_{ju})$. Therefore (6) can be written more compactly as

$$\boldsymbol{X} = \boldsymbol{\Phi}(\boldsymbol{t})\tilde{\boldsymbol{\alpha}} + \boldsymbol{\beta}(\boldsymbol{t}) + \boldsymbol{e}, \tag{7}$$

where $\boldsymbol{\Phi}(\boldsymbol{t}) = \mathrm{diag}(\boldsymbol{\Phi}_1(\boldsymbol{t}_1), \ldots, \boldsymbol{\Phi}_p(\boldsymbol{t}_p))$ with $\boldsymbol{\Phi}_j(\boldsymbol{t}_j) = (\phi_{jv}(t_{ju}))_{u=1,v=1}^{m_j,K_j}$.

**Assumption 6.** *(Sufficient sampling locations) The basis matrix $\boldsymbol{\Phi}(\boldsymbol{t})$ of size $\sum_{j=1}^{p} m_j \times \sum_{j=1}^{p} K_j$ has a full column rank.*

This assumption implies enough sampling locations, over which each random function $Y_j$ is observed, to capture all the causal information that $Y_j$ contains.

Given these six assumptions, our main theorem establishes the causal identifiability of the proposed model.

**Theorem 2.1.** *Under Assumptions 1–6, $\mathcal{S} = (\mathcal{G}, \mathbb{P})$ is causally identifiable.*

The proof essentially involves two steps as shown in Figure 1. On the left-hand side (LHS) of the diagram, we depict the hypergraph-like structure that emerges when assuming the existence of disjoint cycles (Assumption 2), whereas, on the right-hand side (RHS), we offer a magnified view of the hypernodes (nodes containing simple directed cycle). Our approach to proving causal identifiability progresses from the LHS to the RHS. That is, we first prove the identifiability of the hypergraph-like structure depicted on the LHS of Figure 1, and then we proceed to establish the identifiability of each simple directed cycle within every hypernode in the hypergraph. The detailed exposition of the proof can be found in Section A of the Supplementary Materials.

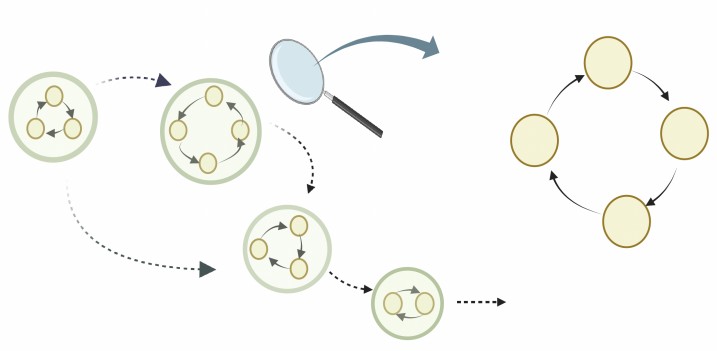

Figure 1: Two important components of causal identifiability proof: (I) identifiability of directed acyclic hypergraph induced by disjoint cycles, and (II) identifiability of each disjoint cycle.

## 3 Bayesian model formulation

In this section, we will describe the inference procedure of the proposed model. A straightforward approach would be a two-step procedure where the first step performs functional principal component analysis on each function marginally to reduce the dimension, and then the second step learns causal structure based on the principal components. However, this simple approach has several disadvantages. First, the estimated functional principal components that explain the most variation of each individual function marginally may not optimally capture the cause-effect dependence relationships among different functions. Second, this procedure is unreliable since estimation uncertainty fails to propagate correctly from the first step to the second step. As such, we propose a fully Bayesian approach, which reduces the dimension of functional data adaptively for causal structure learning.

### 3.1 Model parameters

Let $\boldsymbol{E} = (E_{j\ell})_{j,\ell=1}^{p}$ denote the adjacency matrix where $E_{j\ell} = 1$ indicates the existence of a directed edge from node $\ell$ to node $j$, and 0 otherwise. Let $\{\phi_k\}_{k=1}^{S}$ be a set of $S$ common unknown basis

functions that approximate each random function $Y_j$ i.e., $Y_j = \sum_{k=1}^S \tilde{\alpha}_{jk}\phi_k$ where $\{\tilde{\alpha}_{jk}\}_{k=1}^S$ denote the set of basis coefficients. Note that $\{\phi_k\}$ is not the basis for the lower-dimensional causal embedded subspace $\mathcal{D}_j$. However, we assume that the first $K_j$ of them actually spans $\mathcal{D}_j$ and our goal is to hunt for them through a properly designed inference procedure. Moreover, according to our assumption, we build our SEM on the first $K_j$ of the basis coefficients $\tilde{\boldsymbol{\alpha}}_j = (\tilde{\alpha}_{j1}, \cdots, \tilde{\alpha}_{jK_j})^\top$. Defining $\bar{\boldsymbol{\alpha}}_j = (\tilde{\alpha}_{j,K_j+1}, \cdots, \tilde{\alpha}_{jS})^\top$ with $\bar{\boldsymbol{\alpha}}_j = \boldsymbol{\gamma}_j$, jointly they can be written as

$$\tilde{\boldsymbol{\alpha}} = \tilde{\boldsymbol{B}}\tilde{\boldsymbol{\alpha}} + \tilde{\boldsymbol{\epsilon}}, \tag{8}$$

where $\tilde{\boldsymbol{\alpha}} = (\tilde{\boldsymbol{\alpha}}_1^\top, \cdots, \tilde{\boldsymbol{\alpha}}_p^\top, \bar{\boldsymbol{\alpha}}_1^\top, \ldots, \bar{\boldsymbol{\alpha}}_p^\top)^\top$, $\tilde{\boldsymbol{\epsilon}} = (\tilde{\boldsymbol{\epsilon}}_1^\top, \cdots, \tilde{\boldsymbol{\epsilon}}_p^\top, \boldsymbol{\gamma}_1^\top, \ldots, \boldsymbol{\gamma}_p^\top)^\top$ with $\tilde{\boldsymbol{\epsilon}}_j = (\tilde{\epsilon}_{j1}, \cdots, \tilde{\epsilon}_{jK_j})^\top$ and $\boldsymbol{\gamma}_j = (\gamma_{j,K_j+1}, \ldots, \gamma_{jS})^\top$. Here $\tilde{\boldsymbol{B}} = \begin{pmatrix} \boldsymbol{B} & \boldsymbol{0} \\ \boldsymbol{0} & \boldsymbol{0} \end{pmatrix}$ where $\boldsymbol{B} = ((\boldsymbol{B}_{j\ell}(a,b))_{a=1,b=1}^{K_j,K_l})_{j,\ell=1}^p$ with $\boldsymbol{B}_{jj} = \boldsymbol{0}$ since we assume the absence of self loops. To carry out inference, we assume $\tilde{\epsilon}_{jk}, \gamma_{jk} \overset{ind}{\sim} \sum_{m=1}^{M_{jk}} \pi_{jkm} N(\mu_{jkm}, \tau_{jkm})$.

## 3.2 Adaptive basis expansion

As the $\phi_k$'s are specifically useful for restricting the original function space for each $Y_j$ to a lower-dimensional causally embedded smooth space of dimension $K_j$, we make the basis $\{\phi_k\}$ adaptive for causal structure learning by further expanding them with known spline basis functions [Kowal et al., 2017], $\phi_k(\cdot) = \sum_{r=1}^R A_{kr} b_r(\cdot)$, where $\boldsymbol{b} = (b_1, \ldots, b_R)^\top$ is the set of fixed cubic B-spline basis functions with equally spaced knots and $\boldsymbol{A}_k = (A_{k1}, \ldots, A_{kR})^\top$ are the corresponding spline coefficients. Since we do not fix $\boldsymbol{A}_k$'s a priori, the basis functions $\phi_k$'s can be learned from data a posteriori and hence are adaptive to both data and causal structure (i.e., the basis functions, the functional data, and the causal graph are dependent in their joint distribution).

## 3.3 Prior specifications

**Prior on spline coefficients.** The prior on $A_k$ is chosen to serve multiple purposes. (i) It sorts the basis functions by decreasing smoothness and therefore helps to identify the spanning set of size $K_j$ for the underlying smooth causally embedded space $\mathcal{D}_j$. (ii) Although not a strict requirement for modelling purpose, it forces $\phi_k$'s to be orthonormal, i.e. $\int \phi_k(\omega)\phi_{k'}(\omega)\,d\omega = I(k = k')$. As such, the orthogonality constraints help eliminate any information overlap between the basis functions, which keeps the total number of necessary basis functions that actually contribute to the causal structure learning to a minimum. (iii) It regularizes the roughness of $\phi_k$'s to prevent overfitting.

For (iii), more specifically, we restrict the roughness of the basis functions $\phi_k(\cdot)$ by assigning a prior that penalizes its second derivatives [Gu, 1992, Wahba, 1978, Berry et al., 2002]:

$$\boldsymbol{A}_k \sim N(\boldsymbol{0}, \lambda_k^{-1}\boldsymbol{\Omega}^-),$$

where $\boldsymbol{\Omega}^-$ is the pseudoinverse of $\boldsymbol{\Omega} = \int \boldsymbol{b}''(t)[\boldsymbol{b}''(t)]^\top\,dt$. Let $\boldsymbol{\Omega} = \boldsymbol{U}\boldsymbol{D}\boldsymbol{U}^\top$ be the singular value decomposition of $\boldsymbol{\Omega}$. Following Wand and Ormerod, 2010, to facilitate computation, we reparameterize $\phi_k(\cdot) = \sum_{r=1}^R \tilde{A}_{kr}\tilde{b}_k(\cdot)$ with $\tilde{\boldsymbol{b}}(\cdot) = (1, t, \boldsymbol{b}^T(\cdot)\tilde{\boldsymbol{U}}\tilde{\boldsymbol{D}}^{-\frac{1}{2}})^\top$ where $\tilde{\boldsymbol{D}}$ is the $(R-2) \times (R-2)$ submatrix of $\boldsymbol{D}$ corresponding to non-zero singular values (note that the rank of $\boldsymbol{\Omega}$ is $R-2$ by definition) and $\tilde{\boldsymbol{U}}$ is the corresponding $R \times (R-2)$ submatrix of $\boldsymbol{U}$. This induces a prior on $\tilde{\boldsymbol{A}}_k$ given by

$$\tilde{\boldsymbol{A}}_k \sim N(\boldsymbol{0}, \boldsymbol{S}_k) \text{ with } \boldsymbol{S}_k = \text{diag}(\infty, \infty, \lambda_k^{-1}, \ldots, \lambda_k^{-1}).$$

In other words, the intercept and the linear term are unpenalized but the non-linear terms are penalized, the degree of which is controlled by $\lambda_k$. In practice, we set the first two diagonal elements of $\boldsymbol{S}_k$ as $10^8$. We constrain the regularization parameters $\lambda_1 > \cdots > \lambda_S > 0$ by putting a uniform prior:

$$\lambda_k \sim \text{Uniform}(L_k, U_k), \ \forall \, k \in [S],$$
$$U_1 = 10^8, L_k = \lambda_{k+1} \ \forall \, k \in [S-1],$$
$$U_k = \lambda_{k-1} \ \forall \, k \in \{2, \ldots, S\}, L_S = 10^{-8},$$

which implies that the smoothness of $\phi_k(\cdot)$ decreases as $k$ gets larger.

**Priors on the adjacency matrix.** We propose to use an independent uniform-Bernoulli prior on each entry $E_{j\ell}$ of $\boldsymbol{E}$, i.e., $E_{j\ell}|\rho \overset{\text{ind}}{\sim} \text{Bernoulli}(\rho)$ and $\rho \sim \text{Uniform}(0,1)$. The marginal distribution of $\boldsymbol{E}$ with $\rho$ integrated out is given by

$$p(\boldsymbol{E}) = \int p(\boldsymbol{E}|\rho)p(\rho)\,d\rho = \text{Beta}\left(\sum_{j\neq\ell}E_{j\ell}+1, \sum_{j\neq\ell}(1-E_{j\ell})+1\right).$$

Now, for example, if $\boldsymbol{E}_0$ denotes the null adjacency matrix and $\boldsymbol{E}_1$ denotes the adjacency matrix with only one edge, then we can see that $p(\boldsymbol{E}_0)/p(\boldsymbol{E}_1) = p^2 - p$. Therefore, an empty graph is favored over a graph with one edge by a factor of $p^2 - p$, and, importantly, this penalty increases with $p$. Thus, the uniform-Bernoulli prior prevents false discoveries and leads to a sparse network by increasing the penalty against additional edges as the dimension $p$ grows.

**Prior on the causal effect matrix.** Now given $\boldsymbol{E}$, we assume an independent spike and slab prior on the entries of $\boldsymbol{B} = (\boldsymbol{B}_{j\ell})_{j,\ell=1}^p$:

$$\boldsymbol{B}_{j\ell}|E_{j\ell} \sim (1-E_{j\ell})MVN(\boldsymbol{B}_{j\ell};\mathbf{0}, s\gamma\boldsymbol{I}_{K_j}, \boldsymbol{I}_{K_\ell}) + E_{j\ell}MVN(\boldsymbol{B}_{j\ell};\mathbf{0}, \gamma\boldsymbol{I}_{K_j}, \boldsymbol{I}_{K_\ell}),$$

where $MVN(\boldsymbol{B}_{j\ell};\mathbf{0}, \gamma\boldsymbol{I}_{K_j}, \boldsymbol{I}_{K_\ell})$ is a matrix-variate normal distribution with row and column co-variance matrices as $\gamma\boldsymbol{I}_{K_j}$ and $\boldsymbol{I}_{K_\ell}$, respectively. We assume a conjugate inverse-gamma prior on the causal effect size, $\gamma \sim \text{InverseGamma}(a_\gamma, b_\gamma)$. We choose $a_\gamma = b_\gamma = 1$. We fix $s = 0.02$ so that when $E_{j\ell} = 0$, $\boldsymbol{B}_{j\ell}$ is negligibly small.

**Priors on the parameters of the Gaussian mixture distribution.** We choose conjugate priors for the parameters of the Gaussian mixture distribution:

$$(\pi_{jk1}, \ldots, \pi_{jkM_{jk}}) \sim \text{Dirichlet}(\beta, \ldots, \beta), \quad \forall\, j \in [p], k \in [S]$$

$$\mu_{jkm} \sim N(a_\mu, b_\mu), \ \tau_{jkm} \sim \text{InverseGamma}(a_\tau, b_\tau), \ \forall\, j \in [p], k \in [S], m \in [M_{jk}]$$

We have fixed values for the hyperparameters, $\beta = 1, a_\mu = 0, b_\mu = 100, a_\tau = b_\tau = 1$.

**Prior on the noise variances.** We assume a conjugate prior for $\sigma_j \sim \text{InverseGamma}(a_\sigma, b_\sigma)$, $\forall\, j \in [p]$. We choose $a_\sigma = b_\sigma = 0.01$.

We simulate posterior samples through Markov chain Monte Carlo (MCMC). Details are given in Section B of the Supplementary Materials. Sensitivity analyses will be conducted to test the hyperparameters including $(a_\gamma, b_\gamma), (a_\tau, b_\tau), (a_\sigma, b_\sigma), s, R, S, M$ and $\beta$.

## 4 Simulation study

**Data generation** The data were simulated according to various combinations of sample size $(n)$, number of nodes $(p)$, and grid size $(m_j = d \ \forall j \in [p])$ where $n \in \{75, 150, 300\}$, $p \in \{20, 40, 60\}$, and $d \in \{125, 250\}$. The grid evenly spans the unit interval $[0, 1]$; the results with unevenly spaced grids are presented in Section C of the Supplementary Materials. The true causal graph $\mathcal{G}$ was generated randomly with edge formation probability $2/p$. Given $\mathcal{G}$, each non-zero block $\boldsymbol{B}_{j\ell}$ of the causal effect matrix was generated from the standard matrix-variate normal distribution. We set the true number of basis functions to be $K = 4$. In order to generate $K = 4$ orthonormal basis functions, we first simulated unnormalized basis functions by expanding them further with 6 cubic B-spline basis functions where the coefficients were drawn from the standard normal distribution and then empirically orthonormalized them. The basis coefficients $\tilde{\boldsymbol{\alpha}}$ were generated following (8) with the exogenous variables $\tilde{\epsilon}_j$ drawn independently from Laplace distribution with location parameter $\mu = 0$ and scale parameter $b = 0.2$. We have also considered other non-Gaussian distributions for the exogenous variables; the corresponding results are provided in Section C of the Supplementary Materials. Finally, noisy observations were simulated following (6) with the signal-to-noise ratio, i.e., the mean value of $|Y_j^{(i)}(t)|/\sigma_j$ across all $i$ and $t$, set to 5. Here, superscript $(i)$ denotes the $i$th sample, where $i \in [n]$.

For the implementation of the proposed FENCE, we fixed the number of mixture components to be 10 and ran MCMC for 5,000 iterations (discarding the first 2,000 iterations as burn-in and retaining every 5th iteration after burn-in). The causal graph $G$ was then estimated by using the median probability model [Barbieri and Berger, 2004], i.e., by thresholding the posterior probability of inclusion at 0.5.

**Methods for comparison.** We compared our method with fLiNG [Zhou et al., 2022b], a recently proposed directed acyclic graph (DAG) for multivariate functional data. Codes for Lee and Li [2022], Yang and Suzuki [2022] are not publicly available. Hence for more comparison, we considered two *ad hoc* two-step approaches. In the first step of both approaches, we obtained the basis coefficients by carrying out functional principal component analysis (fPCA) using the `fdapace` [Zhou et al., 2022c] package in R. Then in the second step, given the basis coefficients, we estimated causal graphs using existing causal discovery methods, (i) LiNGAM [Shimizu et al., 2006] (ii) PC [Spirtes and Glymour, 1991] and (iii) CCD [Richardson, 1996]; we call these three approaches fPCA-LiNGAM, fPCA-PC and fPCA-CCD respectively. Note that we did not use SEM-based cyclic discovery algorithm, LiNG-D [Lacerda et al., 2008] in the second step due to the unavailability of the code.

LiNGAM estimates a causal DAG based on the linear non-Gaussian assumption whereas PC generally returns only an equivalence class of DAGs based on conditional independence tests. CCD algorithm is a constraint-based causal discovery method, which yields an equivalence class of cyclic causal graphs. LiNGAM and PC are implemented in the `pcalg` package [Kalisch et al., 2018] in R. CCD algorithm is implemented in the `py-tetrad` [Ramsey et al., 2018] package in python.

**Performance metrics.** To assess the graph recovery performance, we calculated the true positive rate (TPR), false discovery rate (FDR), and Matthew's correlation coefficient (MCC). For TPR and MCC, higher is better, whereas lower FDR is better.

**Results.** Table 1 summarizes the results of 50 repeat simulations, demonstrating that the proposed FENCE model outperforms all competitors (fLiNG, fPCA-LiNGAM, and fPCA-CCD) across all combinations of $n$, $p$, and $d$. We provide the results of fPCA-PC in Section D of the Supplementary Materials, which are similar to those of fPCA-LiNGAM. We favored fPCA-PC and fPCA-CCD by counting a non-invariant edge between two nodes as a true positive as long as the two nodes are adjacent in the true graph. The superiority of FENCE is not unexpected for three reasons. First, fLiNG, fPCA-LiNGAM, and fPCA-PC are not specifically designed for learning cyclic graphs. Second, two-step approaches like fPCA-LiNGAM, fPCA-PC, and fPCA-CCD do not necessarily capture the causally embedded space through the functional principal components. Third, although fPCA-CCD can handle cyclic graphs, it, being a two-step approach, fails to capture true functional dependencies. Overall, these findings provide strong evidence of the effectiveness of FENCE compared to existing methods.

**Additional simulations.** We considered additional simulation scenarios with unevenly spaced grids, general exogenous variable distributions, true acyclic graphs and data generated using non-linear SEM, and also conducted sensitivity analyses of FENCE with respect to several hyperparameters; the results are presented in Section C of the Supplementary Materials. The performance of the proposed method is consistently better than competing methods and is relatively robust with respect to hyperparameter choices.

Table 1: Comparison of performance of various methods under 50 replicates. Since LiNGAM is not applicable to cases where $q > n$ with $q = Kp$ being the total number of extracted basis coefficients across all functions, the results from those cases are not available and indicated by "-". The metrics reported are based on 50 repetitions are reported; standard deviations are given within the parentheses.

| n | p | d | FENCE | | | fLiNG | | | fPCA-LINGAM | | | fPCA-CCD | | |
|---|---|---|---|---|---|---|---|---|---|---|---|---|---|---|
| | | | TPR | FDR | MCC | TPR | FDR | MCC | TPR | FDR | MCC | TPR | FDR | MCC |
| 75 | 20 | 125 | **0.85(0.09)** | **0.19(0.07)** | **0.88(0.05)** | 0.41(0.09) | 0.79(0.05) | 0.36(0.04) | 0.35(0.19) | 0.84(0.04) | 0.11(0.08) | 0.69(0.03) | 0.41(0.04) | 0.23(0.03) |
| 75 | 40 | 125 | **0.79(0.08)** | **0.23(0.06)** | **0.86(0.04)** | 0.37(0.08) | 0.82(0.06) | 0.33(0.05) | - | - | - | 0.73(0.02) | 0.47(0.04) | 0.21(0.05) |
| 75 | 60 | 125 | **0.75(0.07)** | **0.27(0.05)** | **0.83(0.04)** | 0.34(0.07) | 0.83(0.06) | 0.32(0.04) | - | - | - | 0.68(0.03) | 0.61(0.05) | 0.19(0.03) |
| 150 | 20 | 125 | **0.88(0.07)** | **0.14(0.06)** | **0.89(0.05)** | 0.45(0.07) | 0.75(0.06) | 0.39(0.05) | 0.28(0.22) | 0.86(0.05) | 0.08(0.09) | 0.71(0.03) | 0.42(0.03) | 0.25(0.04) |
| 150 | 40 | 125 | **0.81(0.07)** | **0.21(0.06)** | **0.87(0.05)** | 0.39(0.06) | 0.79(0.05) | 0.37(0.04) | 0.35(0.22) | 0.91(0.02) | 0.08(0.06) | 0.73(0.04) | 0.47(0.05) | 0.23(0.03) |
| 150 | 60 | 125 | **0.79(0.06)** | **0.24(0.05)** | **0.86(0.04)** | 0.36(0.04) | 0.80(0.06) | 0.36(0.05) | - | - | - | 0.72(0.05) | 0.54(0.04) | 0.22(0.02) |
| 300 | 20 | 125 | **0.91(0.03)** | **0.09(0.04)** | **0.90(0.04)** | 0.51(0.04) | 0.73(0.06) | 0.41(0.04) | 0.30(0.19) | 0.84(0.05) | 0.11(0.09) | 0.81(0.03) | 0.39(0.04) | 0.26(0.03) |
| 300 | 40 | 125 | **0.87(0.04)** | **0.15(0.05)** | **0.87(0.05)** | 0.47(0.05) | 0.75(0.06) | 0.38(0.05) | 0.27(0.20) | 0.91(0.02) | 0.08(0.06) | 0.77(0.03) | 0.45(0.02) | 0.24(0.03) |
| 300 | 60 | 125 | **0.85(0.05)** | **0.17(0.03)** | **0.86(0.03)** | 0.45(0.05) | 0.76(0.04) | 0.38(0.03) | 0.28(0.17) | 0.91(0.05) | 0.05(0.03) | 0.72(0.03) | 0.49(0.02) | 0.22(0.03) |
| 75 | 20 | 250 | **0.81(0.04)** | **0.23(0.02)** | **0.85(0.05)** | 0.39(0.07) | 0.80(0.05) | 0.39(0.04) | 0.32(0.14) | 0.82(0.03) | 0.09(0.04) | 0.67(0.03) | 0.46(0.03) | 0.22(0.04) |
| 75 | 40 | 250 | **0.73(0.04)** | **0.28(0.05)** | **0.82(0.04)** | 0.35(0.04) | 0.85(0.06) | 0.33(0.05) | - | - | - | 0.68(0.02) | 0.51(0.04) | 0.21(0.03) |
| 75 | 60 | 250 | **0.67(0.03)** | **0.34(0.05)** | **0.79(0.04)** | 0.34(0.04) | 0.85(0.03) | 0.31(0.04) | - | - | - | 0.63(0.04) | 0.56(0.04) | 0.19(0.04) |
| 150 | 20 | 250 | **0.83(0.06)** | **0.17(0.05)** | **0.86(0.05)** | 0.46(0.07) | 0.73(0.07) | 0.42(0.05) | 0.32(0.19) | 0.79(0.05) | 0.13(0.05) | 0.73(0.04) | 0.43(0.02) | 0.24(0.03) |
| 150 | 40 | 250 | **0.79(0.02)** | **0.26(0.06)** | **0.82(0.03)** | 0.41(0.05) | 0.71(0.05) | 0.40(0.03) | 0.31(0.14) | 0.81(0.02) | 0.13(0.06) | 0.71(0.03) | 0.47(0.04) | 0.23(0.03) |
| 150 | 60 | 250 | **0.69(0.05)** | **0.31(0.05)** | **0.79(0.04)** | 0.43(0.03) | 0.79(0.06) | 0.43(0.05) | - | - | - | 0.69(0.02) | 0.52(0.03) | 0.21(0.02) |
| 300 | 20 | 250 | **0.86(0.02)** | **0.16(0.04)** | **0.85(0.04)** | 0.68(0.02) | 0.77(0.07) | 0.47(0.04) | 0.45(0.13) | 0.86(0.05) | 0.17(0.09) | 0.78(0.04) | 0.44(0.06) | 0.27(0.03) |
| 300 | 40 | 250 | **0.79(0.08)** | **0.16(0.05)** | **0.84(0.06)** | 0.73(0.05) | 0.71(0.06) | 0.43(0.05) | 0.39(0.16) | 0.87(0.05) | 0.16(0.04) | 0.76(0.05) | 0.49(0.06) | 0.23(0.05) |
| 300 | 60 | 250 | 0.76(0.05) | **0.21(0.03)** | **0.80(0.03)** | **0.77(0.05)** | 0.74(0.03) | 0.42(0.03) | 0.28(0.17) | 0.90(0.04) | 0.13(0.04) | 0.72(0.06) | 0.53(0.03) | 0.22(0.04) |

# 5 Real data application

**Brain EEG data.** We demonstrate the proposed FENCE model on a brain EEG dataset from an alcoholism study [Zhang et al., 1995]. This dataset was earlier used to demonstrate functional undirected graphical models [Zhu et al., 2016, Qiao et al., 2019] and functional Bayesian network [Zhou et al., 2022b]. Data were initially obtained from 64 electrodes placed on subjects' scalps, which captured EEG signals at 256 Hz (3.9 ms epoch) during a one-second period. The study consists of 122 subjects, out of which 77 are in the alcoholic group and 45 are in the control group. Each subject completed 120 trials. During each trial, the subject was exposed to either a single stimulus (a single picture) or two stimuli (a pair of pictures) shown on a computer monitor. We particularly focus on the EEG signals filtered at $\alpha$ frequency bands between 8 and 12.5Hz using the `eegfilt` function of the `eeglab` toolbox of Matlab as $\alpha$ band signals are associated with inhibitory control [Knyazev, 2007]. Given that the EEG measurements were recorded from each subject over multiple trials, these measurements are not independent of each other due to the time dependency of the trials. Moreover, since the measurements were obtained under various stimuli, the signals may have been affected by different stimulus effects. To mitigate these issues, we calculated the average of the band-filtered EEG signals for each subject across all trials under a single stimulus, resulting in a single event-related potential curve per electrode per subject. By doing so, we eliminated the potential dependence between the measurements and the influence of different stimulus types. We performed separate analyses of the two groups to identify both the similarities and dissimilarities in their brain effective connectivity.

We conducted a Shapiro-Wilk normality test on the observed functions for each of the $p = 64$ scalp positions at each of the $m_j = 256 \ \forall j \in [p]$ time points to evaluate their Gaussianity. The results showed that for numerous combinations of scalp position and time point, the null hypothesis (which assumes that the observations are marginally Gaussian) was rejected. Thus, we conclude that the non-Gaussianity of the proposed model is appropriate. Next, for posterior inference, we ran MCMC for 20,000 iterations, discarded the first half as burn-in, and retained every 10th iteration after burn-in. The estimated causal graph by thresholding the posterior inclusion probability to 0.9 is given below in Figure 2.

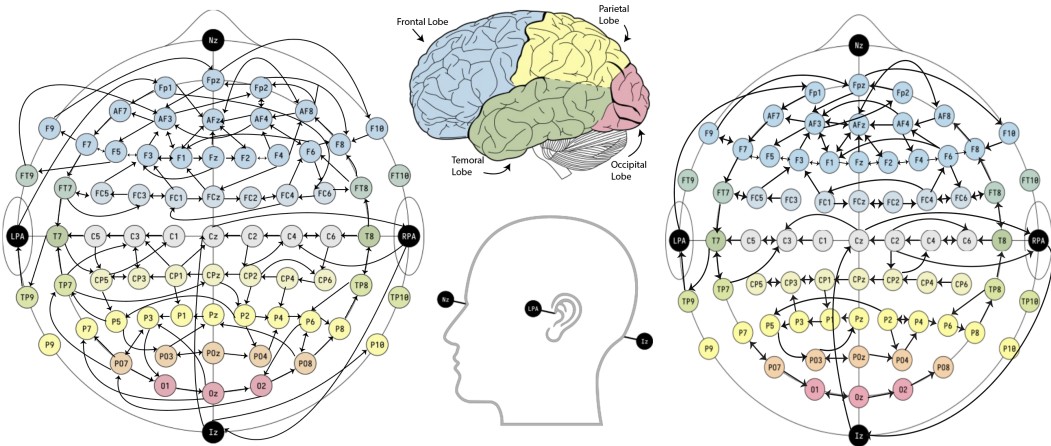

Figure 2: Estimated causal brain connectivity from EEG records by FENCE with posterior probability of inclusion $\geq 0.9$, separately for the alcoholic (left) and control (right) group. The bi-directed edges are just directed cycles, i.e., $i \leftrightarrow j$ means $i \rightarrow j$ and $i \leftarrow j$.

**Results.** There are some interesting findings. First, for both groups (alcoholic and control), brain regions that are located in adjacent positions tend to be more connected than the brain regions that are far apart. Second, dense connectivity is observed in the frontal region of the brain in both groups, with multiple cycles being formed. Third, compared to the control group, the alcoholic group has more connectivity across the left parietal and occipital lobes. Fourth, the same cycle of Iz, Cz, and RPA is observed in both groups.

**Validity.**   We now discuss the validity of our real data results. In Hayden et al., 2007, it was observed that alcohol-dependent subjects exhibited frontal asymmetry, distinguishing them from the control group. Our own investigation aligns well with these results, as we have identified denser connectivity across various brain regions in the middle and left areas of the frontal lobe among alcoholic subjects, when compared to controls. Furthermore, Winterer et al., 2003 documented coherent differences between alcoholics and controls in the posterior hemispheres, specifically in the temporal, parietal, and occipital lobes. In accordance with their findings, our study provides additional support for this claim, as we have observed heightened activity with several cycles formed in those same regions within the alcoholic group when compared to the control group.

## 6   Discussion

We briefly highlight here several potential avenues for future development of our current work. First, an intriguing and important direction would be to explore the relaxation of the causal sufficiency assumption in the model identifiability. Second, our current model is based on a linear non-Gaussian assumption over the exogenous variables, but a nonlinear model could be considered as an alternative. Lastly, an alternative approach to determining the effective number of basis functions that span the lower-dimensional causal embedded space would be to utilize the ordered shrinkage priors [Bhattacharya and Dunson, 2011, Legramanti et al., 2020] in order to adaptively eliminate redundant components, resulting in a more flexible methodology.

## Acknowledgements

Ni's research was partially supported by NSF DMS-2112943 and NIH 1R01GM148974-01.

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
