# Supplementary Materials for "Directed Cyclic Graph for Causal Discovery from Multivariate Functional Data"

## A  Proof of theorem 2.1

*Proof.* For some basis $\{\phi_{jk}\}_{k=1}^{K_j}$ that spans the low dimensional causal embedded space $\mathcal{D}_j$, $\alpha_j$ in (5) of the main manuscript can be further expanded by

$$\alpha_j(t_{ju}) = \sum_{k=1}^{K_j} \tilde{\alpha}_{jk}\phi_{jk}(t_{ju})$$

Using the above , (5) can then be expressed as,

$$X_{ju} = \sum_{k=1}^{K_j} \tilde{\alpha}_{jk}\phi_{jk}(t_{ju}) + \beta_j(t_{ju}) + e_{ju}, \forall j \in [p], u \in [m_j] \tag{1}$$

More compactly, the above (1) can be rewritten as,

$$\boldsymbol{X} = \boldsymbol{\Phi(t)}\tilde{\boldsymbol{\alpha}} + \boldsymbol{\beta(t)} + \boldsymbol{e} \tag{2}$$

where $\boldsymbol{X} = (\boldsymbol{X}_1^\top, \ldots, \boldsymbol{X}_p^\top)^\top, \tilde{\boldsymbol{\alpha}} = (\tilde{\boldsymbol{\alpha}}_1^\top, \ldots, \tilde{\boldsymbol{\alpha}}_p^\top)^\top, \boldsymbol{\beta(t)} = (\boldsymbol{\beta}_1(\boldsymbol{t}_1)^\top, \ldots, \boldsymbol{\beta}_p(\boldsymbol{t}_p)^\top)^\top, \boldsymbol{e} = (\boldsymbol{e}_1^\top, \ldots, \boldsymbol{e}_p^\top)^\top$ and $\boldsymbol{\Phi(t)} = \text{diag}(\boldsymbol{\Phi}_1(\boldsymbol{t}_1), \ldots, \boldsymbol{\Phi}_p(\boldsymbol{t}_p))$ with $\boldsymbol{X}_j = (X_{j1}, \ldots, X_{jm_j})^\top, \tilde{\boldsymbol{\alpha}}_j = (\tilde{\alpha}_{j1}, \ldots, \tilde{\alpha}_{jK_j})^\top, \boldsymbol{\beta}_j(\boldsymbol{t}_j) = (\beta_j(t_{j1}), \ldots, \beta_j(t_{jm_j}))^\top, \boldsymbol{e}_j = (e_{j1}, \ldots, e_{jm_j})^\top$ and

$$\boldsymbol{\Phi}_j(\boldsymbol{t}_j) = \begin{pmatrix} \phi_{j1}(t_{j1}) & \phi_{j2}(t_{j1}) & \cdots & \phi_{jK_j}(t_{j1}) \\ \phi_{j1}(t_{j2}) & \phi_{j2}(t_{j2}) & \cdots & \phi_{jK_j}(t_{j2}) \\ \vdots & \vdots & \ddots & \vdots \\ \phi_{j1}(t_{jm_j}) & \phi_{j2}(t_{jm_j}) & \cdots & \phi_{jK_j}(t_{jm_j}) \end{pmatrix}$$

The structural equation model is then defined on $\tilde{\boldsymbol{\alpha}}$ as,

$$\tilde{\boldsymbol{\alpha}} = \boldsymbol{B}\tilde{\boldsymbol{\alpha}} + \tilde{\boldsymbol{\epsilon}}$$
$$\Rightarrow \tilde{\boldsymbol{\alpha}} = \boldsymbol{\Omega}\tilde{\boldsymbol{\epsilon}}, \ [\text{by Assumption 3}] \tag{3}$$

where $\boldsymbol{\Omega} = (\boldsymbol{I} - \boldsymbol{B})^{-1}$.

Referring to Assumption 5 of section 2.3 in the main manuscript, we write $\boldsymbol{\beta(t)} = \boldsymbol{C(t)}\boldsymbol{\gamma}$ where $\gamma_{jk} \sim \sum_{m=1}^{M_{jk}} \pi'_{jkm} N(\mu'_{jkm}, \tau'_{jkm})$ with

$$\boldsymbol{C} = \begin{pmatrix} \boldsymbol{C}_{11}(\boldsymbol{t}_1) & \boldsymbol{0} & \cdots & \boldsymbol{0} \\ \boldsymbol{0} & \boldsymbol{C}_{22}(\boldsymbol{t}_2) & \cdots & \boldsymbol{0} \\ \vdots & \vdots & \ddots & \vdots \\ \boldsymbol{0} & \boldsymbol{0} & \cdots & \boldsymbol{C}_{pp}(\boldsymbol{t}_p) \end{pmatrix}$$

Using this representation for $\boldsymbol{\beta(t)}$ and (3), (2) boils down to,

$$\boldsymbol{X} = \boldsymbol{\Phi(t)}\boldsymbol{\Omega}\tilde{\boldsymbol{\epsilon}} + \boldsymbol{C(t)}\boldsymbol{\gamma} + \boldsymbol{e} \tag{4}$$

From here on let us define $N = \sum_{j=1}^{p} m_j$ and $K = \sum_{j=1}^{p} K_j$. We define two class variables $\boldsymbol{\xi}$ and $\boldsymbol{\eta}$ such that $\epsilon_{jk}|\xi_{jk} = m \sim N(\mu_{jkm}, \tau_{jkm})$ and $\mathbb{P}(\xi_{jk} = m) = \pi_{jkm}$ and $\gamma_{jk}|\eta_{jk} = m \sim N(\mu'_{jkm}, \tau'_{jkm})$ and $\mathbb{P}(\eta_{jk} = m) = \pi'_{jkm}$. Conditioning on these class variables $\boldsymbol{\xi}$ and $\boldsymbol{\eta}$,

$$\boldsymbol{X}|\boldsymbol{\xi}, \boldsymbol{\eta} \sim N(\boldsymbol{\mu_X}, \boldsymbol{\Sigma_X}) \tag{5}$$

where,

$$\boldsymbol{\mu_X} = \boldsymbol{\Phi(t)}\boldsymbol{\Omega}\boldsymbol{\mu_\xi} + \boldsymbol{C(t)}\boldsymbol{\mu_\eta}$$
$$\boldsymbol{\Sigma_X} = \boldsymbol{\Phi(t)}\boldsymbol{\Omega}\boldsymbol{T_\xi}\boldsymbol{\Omega}^\top\boldsymbol{\Phi(t)}^\top + \boldsymbol{C(t)}\boldsymbol{T_\eta}\boldsymbol{C(t)}^\top + \boldsymbol{\Sigma}$$

13  with $\boldsymbol{\mu_\xi} = (\boldsymbol{\mu_{\xi_1}^\top}, \ldots, \boldsymbol{\mu_{\xi_p}^\top})^\top$ and $\boldsymbol{\mu_\eta} = (\boldsymbol{\mu_{\eta_1}^\top}, \ldots, \boldsymbol{\mu_{\eta_p}^\top})^\top$ are the collection of means and $\boldsymbol{T_\xi} =$
14  $\mathrm{diag}(\boldsymbol{T_{\xi_1}}, \ldots, \boldsymbol{T_{\xi_p}})$ and $\boldsymbol{T_\eta} = \mathrm{diag}(\boldsymbol{T_{\xi_1}}, \ldots, \boldsymbol{T_{\xi_p}})$ are diagonal matrices with variances as diagonal
15  entries corresponding to the class variable $\boldsymbol{\xi}$ and $\boldsymbol{\eta}$. Here, $\boldsymbol{\mu_{\xi_j}} = (\mu_{\xi_{j1}}, \ldots, \mu_{\xi_{jK_j}})^\top, \boldsymbol{\mu_{\eta_j}} =$
16  $(\mu_{\eta_{j1}}, \ldots, \mu_{\eta_{jK_j}})^\top, \boldsymbol{T_{\xi_j}} = \mathrm{diag}(T_{\xi_{j1}}, \ldots, T_{\xi_{jK_j}})$ and $\boldsymbol{T_{\eta_j}} = \mathrm{diag}(T_{\eta_{j1}}, \ldots, T_{\eta_{ju}}, \ldots)$ with $\mu_{\xi_{jk}} =$
17  $\mu_{jkm}$ if $\xi_{jk} = m$ and $T_{\xi_{jk}} = \tau_{jkm}$ if $\xi_{jk} = m$ and $\mu_{\eta_{jk}} = \mu'_{jkm}$ if $\eta_{jk} = m$ and $T_{\eta_{jk}} = \tau'_{jkm}$ if
18  $\eta_{jk} = m$. $\boldsymbol{\Sigma}^{N \times N} = \mathrm{diag}(\sigma_1, \ldots, \sigma_1, \ldots, \sigma_p, \ldots, \sigma_p)$.

19  Our causal identifiability proof necessarily involves **two steps** - First, we shall prove that the
20  hypergraph like structure which is formed under the assumption of existence of disjoint cycles
21  (Refer Assumption 2 of the main manuscript) is identifiable. Second, the disjoint cycles inside every
22  hypernode are identifiable. Please refer to Figure 1 in the main paper for an artistic exposition of the
23  proof structure.

24  **Step 1.**  Now we shall prove the identifiability of our model under the assumption that the SEM
25  involving $\tilde{\boldsymbol{\alpha}}$ has an underlying graph in which the cycles are disjoint (Assumption 2). Under this
26  assumption, we will have cycles of variable length which are connected by directed edges such that
27  no two cycles in the graph have two nodes that are common to both. This induces a hypergraph like
28  structure with each disjoint cycle forming a simple directed cycle in $\mathcal{V}$.

29  Let us mathematically formalize what we have discussed in the above paragraph. Suppose, $\mathfrak{C} =$
30  $\{\mathcal{C}_1, \ldots, \mathcal{C}_u\}$ where each $\mathcal{C}_i$ is a simple directed cycle. Clearly, $\mathcal{V} = \cup_{i=1}^u \mathcal{C}_i$ as $\mathcal{C}_i$s form a partition
31  in $\mathcal{V}$. Without loss of generality, let us assume that $\{\tilde{\boldsymbol{\alpha}}_1, \ldots, \tilde{\boldsymbol{\alpha}}_p\}$ be arranged in such a way that the
32  first $r_1$ elements form the simple cycle $\mathcal{C}_1$, the next $r_2$ elements form another simple cycle $\mathcal{C}_2$ and
33  so on such that $\sum_{i=0}^u r_i = p$ with $r_0 = 0$ and $\mathcal{C}_i = \{\tilde{\boldsymbol{\alpha}}_{r_{i-1}+1}, \ldots, \tilde{\boldsymbol{\alpha}}_{r_i}\}$. We denote the hypergraph
34  formed by $\mathfrak{C}$ by $\bar{\mathcal{G}}$.

35  Let $\bar{\mathcal{G}}$ and $\bar{\mathcal{G}}'$ be two graphs where $\bar{\mathcal{G}}' \neq \bar{\mathcal{G}}$. We can assume a topological ordering in $\bar{\mathcal{G}}$ in a sense
36  that if $\mathcal{C}_q \to \mathcal{C}_r$ then $q < r$. Therefore, the $\boldsymbol{B}$ induced by the graph $\bar{\mathcal{G}}$ is necessarily a lower block
37  triangular matrix with block $\boldsymbol{0}$ as the diagonal entries. We cannot say any such thing about the matrix
38  $\boldsymbol{B}'$ induced by the graph $\bar{\mathcal{G}}'$ except that having block $\boldsymbol{0}$ matrices as it's diagonal elements.

39  Let $\mathbb{P}$ and $\mathbb{P}'$ be the joint probability distribution of $\boldsymbol{X}$ associated with the two graphs $\mathcal{G}$ and $\mathcal{G}'$
40  respectively. Let $\mathcal{S} = (\bar{\mathcal{G}}, \mathbb{P})$ and $\mathcal{S}' = (\bar{\mathcal{G}}', \mathbb{P}')$. We shall prove by contradiction that $\mathcal{S}$ and $\mathcal{S}'$ are
41  not equivalent.

42  Suppose, $\mathbb{P}(\boldsymbol{X}) \equiv \mathbb{P}'(\boldsymbol{X})$. Then due to the identifiability of finite Gaussian mixture models up to
43  label permutation [Teicher, 1963, Yakowitz and Spragins, 1968], we must have, for any $\boldsymbol{\xi}, \boldsymbol{\eta}$,

$$\boldsymbol{\Phi}(t)\boldsymbol{\Omega T_\xi \Omega}^\top \boldsymbol{\Phi}(t)^\top + \boldsymbol{C}(t)\boldsymbol{T_\eta C}(t)^\top + \boldsymbol{\Sigma} = \boldsymbol{\Phi}(t)\boldsymbol{\Omega}' \boldsymbol{T'_\xi \Omega'}^\top \boldsymbol{\Phi}(t)^\top + \boldsymbol{C}(t)\boldsymbol{T'_\eta C}(t)^\top + \boldsymbol{\Sigma}' \quad (6)$$

44  For some choice of $\tilde{\boldsymbol{\xi}} \neq \boldsymbol{\xi}$ and $\tilde{\boldsymbol{\eta}} = \boldsymbol{\eta}$, we can write from (6),

$$\boldsymbol{\Phi}(t)\boldsymbol{\Omega}(\boldsymbol{T_\xi} - \boldsymbol{T_{\tilde{\xi}}})\boldsymbol{\Omega}^\top \boldsymbol{\Phi}(t)^\top = \boldsymbol{\Phi}(t)\boldsymbol{\Omega}'(\boldsymbol{T'_\xi} - \boldsymbol{T'_{\tilde{\xi}}})\boldsymbol{\Omega'}^\top \boldsymbol{\Phi}(t)^\top$$
$$\Rightarrow \boldsymbol{\Omega}(\boldsymbol{T_\xi} - \boldsymbol{T_{\tilde{\xi}}})\boldsymbol{\Omega}^\top = \boldsymbol{\Omega}'(\boldsymbol{T'_\xi} - \boldsymbol{T'_{\tilde{\xi}}})\boldsymbol{\Omega'}^\top, \text{(using Assumption 6)} \quad (7)$$

45  Notice that $\boldsymbol{\Omega}$ being an invertible matrix, every row of every block diagonal matrices must have at
46  least a non zero element. $\boldsymbol{\Omega}_{K.}$ denotes the last row for $\boldsymbol{\Omega}$ and $l_1$ be the extreme position for which
47  $\boldsymbol{\Omega}_{K,l_1} \neq 0$. Pick $\tilde{\boldsymbol{\xi}}$ above such that $\tilde{\boldsymbol{\xi}} = \boldsymbol{\xi}$ except for that $l_1$th element such that $(\boldsymbol{T_\xi} - \boldsymbol{T_{\tilde{\xi}}})_{l_1,l_1} \neq$
48  $0$. Hence the matrix $(\boldsymbol{T_\xi} - \boldsymbol{T_{\tilde{\xi}}})$ is of rank 1 and from (7) it implies that $\exists \, s_1 \in [K]$ such that
49  $(\boldsymbol{T'_\xi} - \boldsymbol{T'_{\tilde{\xi}}})_{s_1,s_1} \neq 0$. Therefore clearly,

$$0 \neq \boldsymbol{\Omega}_{K,.}(\boldsymbol{T_\xi} - \boldsymbol{T_{\tilde{\xi}}})\boldsymbol{\Omega}_{K,.}^T = \boldsymbol{\Omega}'_{K,.}(\boldsymbol{T'_\xi} - \boldsymbol{T'_{\tilde{\xi}}})\boldsymbol{\Omega'}_{K,.}^\top = \boldsymbol{\Omega}'^2_{K,s_1}(\boldsymbol{T'_\xi} - \boldsymbol{T'_{\tilde{\xi}}})_{s_1,s_1} \quad (8)$$

50  Now as $(\boldsymbol{T'_\xi} - \boldsymbol{T'_{\tilde{\xi}}})_{s_1,s_1} \neq 0$, we have from (8), $\boldsymbol{\Omega}'_{K,s_1} \neq 0$. Similarly, if we now focus on the
51  $(K-1)$th row of $\boldsymbol{\Omega}$, there can be two cases,

52  **Case 1:** The last position for which $\boldsymbol{\Omega}_{K-1,.} \neq 0$ coincides with $l_1$. Then for this, we shall proceed
53  with the same choice of $\tilde{\boldsymbol{\xi}}$ as above and with the same argument from above we can show that
54  $\boldsymbol{\Omega}'_{K-1,s_1} \neq 0$

55 **Case 2:** If the position of the last non zero element in the $(K-1)$th row of $\mathbf{\Omega}$ is some $l_2 (\neq l_1)$,
56 we pick $\tilde{\boldsymbol{\xi}}$ such that $\tilde{\boldsymbol{\xi}} = \boldsymbol{\xi}$ except for that $l_2$th element such that $(\boldsymbol{T_\xi} - \boldsymbol{T_{\tilde{\xi}}})_{l_2, l_2} \neq 0$. Hence the
57 matrix $(\boldsymbol{T_\xi} - \boldsymbol{T_{\tilde{\xi}}})$ is of rank 1 and from (7) it implies that $\exists\, s_2 \in [K]$ such that $(\boldsymbol{T'_\xi} - \boldsymbol{T'_{\tilde{\xi}}})_{s_2, s_2} \neq 0$.
58 Therefore clearly,

$$0 \neq \mathbf{\Omega}_{K-1,.}(\boldsymbol{T_\xi} - \boldsymbol{T_{\tilde{\xi}}})\mathbf{\Omega}_{K-1,.}^T = \mathbf{\Omega}'_{K-1,.}(\boldsymbol{T'_\xi} - \boldsymbol{T'_{\tilde{\xi}}})\mathbf{\Omega}'^\top_{K-1,.} = \mathbf{\Omega}'^2_{K-1,s_2}(\boldsymbol{T'_\xi} - \boldsymbol{T'_{\tilde{\xi}}})_{s_2, s_2} \tag{9}$$

59 Similarly as before since $(\boldsymbol{T'_\xi} - \boldsymbol{T'_{\tilde{\xi}}})_{s_2, s_2} \neq 0$, we have from (9), $\mathbf{\Omega}'_{K-1,s_2} \neq 0$. Define $K_{|\mathcal{C}_i|} =$
60 $\sum_{j=r_{i-1}+1}^{r_i} K_j$. Clearly, $\sum_{i=1}^{u} \sum_{j=r_{i-1}+1}^{r_i} K_j = K$. Therefore, proceeding similarly from above
61 we can show that $\mathbf{\Omega}'_{K-K_{|\mathcal{C}_u|}+1, s_{K_{|\mathcal{C}_u|}}} \neq 0$.

62 Now since $\mathbf{\Omega}$ is a lower block triangular matrix, we have $\forall\, r \leq K - K_{|\mathcal{C}_u|}, \mathbf{\Omega}_{r,(K-K_{|\mathcal{C}_u|}+1):K} = 0$.
63 Therefore, if we pick some $\tilde{\boldsymbol{\xi}}$ which does not match $\boldsymbol{\xi}$ at the $l_j$th position, $l_j > K - K_{|\mathcal{C}_u|}, j \in [K_{|\mathcal{C}_u|}]$
64 such that $(\boldsymbol{T_\xi} - \boldsymbol{T_{\tilde{\xi}}})_{l_j, l_j} \neq 0$ then there will exist some $s_j \in [K]$ such that $(\boldsymbol{T'_\xi} - \boldsymbol{T'_{\tilde{\xi}}})_{s_j, s_j} \neq 0, j \in$
65 $[K_{|\mathcal{C}_u|}]$. Therefore we have,

$$0 = \mathbf{\Omega}_{r,.}(\boldsymbol{T_\xi} - \boldsymbol{T_{\tilde{\xi}}})\mathbf{\Omega}_{r,.}^T = \mathbf{\Omega}'_{r,.}(\boldsymbol{T'_\xi} - \boldsymbol{T'_{\tilde{\xi}}})\mathbf{\Omega}'^\top_{r,.} = \mathbf{\Omega}'^2_{r,s_j}(\boldsymbol{T'_\xi} - \boldsymbol{T'_{\tilde{\xi}}})_{s_j, s_j} \tag{10}$$

66 From (10), as $(\boldsymbol{T'_\xi} - \boldsymbol{T'_{\tilde{\xi}}})_{s_j, s_j} \neq 0$, we have, $\mathbf{\Omega}'_{r,s_j} = 0, \forall\, r \leq K - K_{|\mathcal{C}_u|}, j \in [K_{|\mathcal{C}_u|}]$.

67 Proceeding similarly from above, if we repeat the above set of arguments for all the rows of $\mathbf{\Omega}$
68 matrix, we can observe that $\mathbf{\Omega}'$ is just a block column permutation of a lower block triangular matrix.
69 Therefore there exists a block lower triangular matrix $\boldsymbol{A}$ and a block permutation matrix $\boldsymbol{P}$ such that,

$$\mathbf{\Omega}' = \boldsymbol{A}\boldsymbol{P}$$
$$\Rightarrow (\boldsymbol{I} - \boldsymbol{B}')^{-1} = \boldsymbol{A}\boldsymbol{P}$$
$$\Rightarrow (\boldsymbol{I} - \boldsymbol{B}') = \boldsymbol{P}^\top \boldsymbol{A}^{-1} \tag{11}$$

70 Now, the RHS of (11) is just a row permuted block lower triangular matrix. Therefore, the permutation
71 matrix $\boldsymbol{P}$ has to be the identity matrix; otherwise $\boldsymbol{P}^T \boldsymbol{A}^{-1}$ must have zeros in its diagonal but $\boldsymbol{I} - \boldsymbol{B}'$
72 has unit diagonal because $\boldsymbol{B}'$ has zero diagonal (no self-loop). Hence we arrive at a contradiction
73 and conclude from here that $\mathcal{S}$ and $\mathcal{S}'$ are not equivalent, i.e. $\mathbb{P}(\boldsymbol{X}) \neq \mathbb{P}'(\boldsymbol{X})$.

74 **Step 2.** We now try to prove that each simple directed cycle is identifiable.

75 If $\boldsymbol{H}$ and $\boldsymbol{H}'$ are the sub-matrices induced by some $\mathcal{C}_j \in \mathfrak{C}, j \in [u]$ in $\bar{\mathcal{G}}$ and $\bar{\mathcal{G}}'$ respectively then it
76 is sufficient to show that for any permutation matrix $\boldsymbol{P}, \boldsymbol{P}(\boldsymbol{I} - \boldsymbol{H}) = \boldsymbol{I} - \boldsymbol{H}' \Rightarrow \boldsymbol{P} = \boldsymbol{I}$.

77 Now since $\boldsymbol{H}$ is a matrix for a simple cycle, it can be written as $\boldsymbol{H} = \boldsymbol{Q}\boldsymbol{D}$ where $\boldsymbol{Q}$ is a permutation
78 matrix and $\boldsymbol{D}$ is block diagonal matrix. Now,

$$\boldsymbol{P}(\boldsymbol{I} - \boldsymbol{H}) = \boldsymbol{I} - \boldsymbol{H}'$$
$$\Rightarrow \boldsymbol{P}(\boldsymbol{I} - \boldsymbol{Q}\boldsymbol{D}) = \boldsymbol{I} - \boldsymbol{H}'$$
$$\Rightarrow \boldsymbol{P} - \boldsymbol{P}\boldsymbol{Q}\boldsymbol{D} = \boldsymbol{I} - \boldsymbol{H}' \tag{12}$$

79 The RHS of (12) has all $1'$s in it's diagonal. Therefore the diagonal elements of $\boldsymbol{P}$ and $\boldsymbol{P}\boldsymbol{Q}$ i.e.
80 $(\boldsymbol{P})_{i,i}$ and $(\boldsymbol{P}\boldsymbol{Q})_{i,i}$ cannot be simultaneously 0.

81 **Case 1:** $(\boldsymbol{P})_{i,i} \neq 1$ for some $i$.

82 Without loss of generality let, $\boldsymbol{P} = \begin{pmatrix} \boldsymbol{P_1} & \boldsymbol{0} \\ \boldsymbol{0} & \boldsymbol{I} \end{pmatrix}$ where $\boldsymbol{P_1}$ is the matrix that has 0 in it's diagonal.

83 Clearly, $\boldsymbol{P_1} = (\boldsymbol{I} \quad \boldsymbol{0}) \boldsymbol{P} \begin{pmatrix} \boldsymbol{I} \\ \boldsymbol{0} \end{pmatrix}$. Now from (12),

$$(\boldsymbol{I} \quad \boldsymbol{0})\,(\boldsymbol{P} - \boldsymbol{P}\boldsymbol{Q}\boldsymbol{D})\begin{pmatrix}\boldsymbol{I}\\\boldsymbol{0}\end{pmatrix} = (\boldsymbol{I} \quad \boldsymbol{0})\,(\boldsymbol{I} - \boldsymbol{H}')\begin{pmatrix}\boldsymbol{I}\\\boldsymbol{0}\end{pmatrix}$$

$$\Rightarrow \boldsymbol{P_1} - (\boldsymbol{P_1} \quad \boldsymbol{0})\,\boldsymbol{Q}\begin{pmatrix}\boldsymbol{D_1}\\\boldsymbol{0}\end{pmatrix} = \boldsymbol{I} - (\boldsymbol{I} \quad \boldsymbol{0})\,\boldsymbol{H}'\begin{pmatrix}\boldsymbol{I}\\\boldsymbol{0}\end{pmatrix}$$

$$\Rightarrow \boldsymbol{P_1} - \boldsymbol{P_1}\boldsymbol{Q_{11}}\boldsymbol{D_1} = \boldsymbol{I} - \boldsymbol{H}'_{11} \tag{13}$$

Notice that, the diagonals of RHS of (13) are equal to 1, $\boldsymbol{Q_{11}}$ is not necessarily a permutation matrix but it has at most one 1 in every column and $\boldsymbol{P_1}$ is a permutation matrix. Following from the same argument as before, we can therefore say that the diagonals of $\boldsymbol{P_1}$ and $\boldsymbol{P_1}\boldsymbol{Q_{11}}$ cannot be simultaneously 0. Now from our assumption since $(\boldsymbol{P_1})_{i,i} = 0, \forall\, i$ we have,

$$(\boldsymbol{P_1}\boldsymbol{Q_{11}})_{i,i} = 1, \forall\, i$$

Now since $\boldsymbol{P_1}$ is a permutation matrix and $\boldsymbol{Q_{11}}$ has at most one 1 in every column, we have $\boldsymbol{P_1}\boldsymbol{Q_{11}} = \boldsymbol{I}$ and $\boldsymbol{D_1} = -\boldsymbol{I}$ Therefore, from above we obtain,

$$\boldsymbol{I} + \boldsymbol{P_1} = \boldsymbol{I} - \boldsymbol{H}'_{11}$$

$$\Rightarrow \boldsymbol{P_1} = -\boldsymbol{H}'_{11} \tag{14}$$

Let any eigenvalue of matrix $\boldsymbol{A}$ be donoted by $\lambda(\boldsymbol{A})$. Therefore from (14), we can obtain,

$$\lambda(\boldsymbol{P_1}) = \lambda(-\boldsymbol{H}'_{11})$$

$$\Rightarrow \lambda(\boldsymbol{P_1}) = -\lambda(\boldsymbol{H}'_{11}), (\because -\lambda \text{ is an eigenvalue for } \boldsymbol{H_{11}})$$

$$\Rightarrow |\lambda(\boldsymbol{P_1})| = |\lambda(\boldsymbol{H}'_{11})|, (\text{taking modulus on both sides})$$

Now since $\boldsymbol{P_1}$ is a permutation matrix, all of it's eigenvalues lie on a unit circle, i.e. $|\lambda(\boldsymbol{P_1})| = 1$. But according to Assumption 3 of the main manuscript, the moduli of the eigenvalues of $\boldsymbol{H}'$ and hence $\boldsymbol{H}'_{11}$ are less than 1 and none of the real eigenvalues are equal to 1. Therefore, we arrive at a contradiction.

**Case 2:** $(\boldsymbol{P})_{i,i} = 0\ \forall i$

Therefore, $(\boldsymbol{P}\boldsymbol{Q})_{i,i} = 1\ \forall i$ and $\boldsymbol{D} = -\boldsymbol{I}$. Therefore from (12), we obtain,

$$\boldsymbol{P} + \boldsymbol{I} = \boldsymbol{I} - \boldsymbol{H}'$$

Proceeding similarly from the case 1 argument, we arrrive at a contradiction.

$$\therefore \boldsymbol{P} = \boldsymbol{I}$$

$\square$

# B   Posterior inference

## B.1   Selecting the effective number of basis functions for the causal embedded space

While it is possible to use a prior to learn the number of basis functions jointly with other parameters through reversible jump MCMC or to use shrinkage priors to adaptively truncate and eliminate redundant functions, these approaches can lead to significant computational burden and potential Markov chain mixing issues. Therefore, this article employs a simple heuristic approach, as described in Kowal et al., 2017, Zhou et al., 2022. First, the functional observations are imputed and arranged into a $(n \times p) \times d$ matrix, where $d = |\cup_{i,j} \mathcal{T}_j^{(i)}|$ represents the size of the union of the measurement grid over all realized random functions. Then, singular value decomposition is performed, and the minimum value of $K$ is selected such that its proportion of variance explained is at least 90%. This value is fixed throughout MCMC. It should be noted that while $K$ remains fixed, the basis functions are adaptively inferred.

We have noted that the value of $K$ derived from the aforementioned heuristic method falls within a range of $\pm 2$ in comparison to the value obtained by fixing a grid encompassing values $\{1, 2, 3, 4, 5, 6, 7\}$ for $K$ and subsequently selecting the $K$ associated with the lowest WAIC [Watanabe, 2013]. The graph recovery performance, as assessed by Matthew's correlation coefficient (MCC) using this method, closely aligns with that of the previous approach. Consequently, we adopted the aforementioned heuristic technique to determine the optimal number of basis functions that collectively span the causal embedded space.

## B.2 Posterior distributions

While the closed form expression for the posterior distribution cannot be obtained, we resort to MCMC techniques for sampling. We use superscript $(\cdot)$ to denote observations throughout the text. Let $\boldsymbol{X}^{(1)}, \ldots, \boldsymbol{X}^{(n)}$ be $n$ realizations of the multivariate random functions $\boldsymbol{X}$. For the mixture of Gaussian distribution we assume $M_{jk} = M$ for simplicity. In order to obtain updates for the parameters of the mixture distribution, we define a class variable $\boldsymbol{\xi}^{(i)} = (\boldsymbol{\xi}_1^{(i)\top}, \ldots, \boldsymbol{\xi}_p^{(i)\top}, \bar{\boldsymbol{\xi}}_1^{(i)\top}, \ldots, \bar{\boldsymbol{\xi}}_p^{(i)\top})^\top$ with $\boldsymbol{\xi}_j^{(i)} = (\xi_{j1}^{(i)}, \ldots, \xi_{jK_j}^{(i)})^\top$ and $\bar{\boldsymbol{\xi}}_j^{(i)} = (\xi_{j,K_j+1}^{(i)}, \ldots, \xi_{jS}^{(i)})^\top$ where $\xi_{jk}^{(i)} = m$ if $\tilde{\epsilon}_{jk}^{(i)}$ belongs to the mixture component $m$. Let $\boldsymbol{M}^{(i)} = (\boldsymbol{\mu}_1^{(i)\top}, \ldots, \boldsymbol{\mu}_p^{(i)\top}, \bar{\boldsymbol{\mu}}_1^{(i)\top}, \ldots, \bar{\boldsymbol{\mu}}_p^{(i)\top})^\top$ and $\boldsymbol{T}^{(i)} = \mathrm{diag}(\boldsymbol{\tau}_1^{(i)}, \ldots, \boldsymbol{\tau}_p^{(i)}, \bar{\boldsymbol{\tau}}_1^{(i)}, \ldots, \bar{\boldsymbol{\tau}}_p^{(i)})$ be the mean and covariance matrix of $\boldsymbol{\epsilon}^{(i)}$ where $\boldsymbol{\mu}_j^{(i)} = (\mu_{j1}^{(i)}, \ldots, \mu_{jK_j}^{(i)})^\top$, $\bar{\boldsymbol{\mu}}_j^{(i)} = (\mu_{j,K_j+1}^{(i)}, \ldots, \mu_{jS}^{(i)})^\top$, $\boldsymbol{\tau}_j^{(i)} = (\tau_{j1}^{(i)}, \ldots, \tau_{jS}^{(i)})^\top$ and $\bar{\boldsymbol{\tau}}_j^{(i)} = (\tau_{j,K_j+1}^{(i)}, \ldots, \tau_{jS}^{(i)})^\top$ with $\mu_{jk}^{(i)} = \sum_{m=1}^M \mu_{jkm} \mathbf{1}(\xi_{jk}^{(i)} = m)$ and $\tau_{jk}^{(i)} = \sum_{m=1}^M \tau_{jkm} \mathbf{1}(\xi_{jk}^{(i)} = m)$. Define $\boldsymbol{\pi}_{jk} = (\pi_{jk1}, \ldots, \pi_{jkm})^\top, \forall j \in [p], k \in [S]$. Let $\tilde{\boldsymbol{\epsilon}}^{(i)} = \tilde{\boldsymbol{\alpha}}^{(i)} - \tilde{\boldsymbol{B}}\tilde{\boldsymbol{\alpha}}^{(i)}$ be the vector of exogenous variables for the $i^{\mathrm{th}}$ observation.

**Posterior distribution of the parameters of the mixture distribution.** For each $j \in [p], k \in [S]$, update the mixture weights $\boldsymbol{\pi}_{jk}$ by drawing from a Dirichlet distribution with concentration parameters $\{\beta_m\}_{m\in[M]}$ where,

$$\beta_m = \alpha + \sum_{i=1}^n \mathbf{1}(\xi_{jk}^{(i)} = m) \tag{15}$$

Now, given the $\boldsymbol{\pi}_{jk}$'s, for each $i \in [n], j \in [p], k \in [S]$, update the class variables $\xi_{jk}^{(i)}$ from a categorical distribution with class probability $\{\pi_m^{(i)}\}_{m\in M}$ where, $\pi_m^{(i)} \propto \pi_{jkm} \mathrm{N}(\tilde{\epsilon}_{jk}^{(i)}; \mu_{jkm}, \tau_{jkm})$ with $\sum_{m=1}^M \pi_m^{(i)} = 1$.

$$\pi_m^{(i)} \propto \pi_{jkm} \mathrm{N}(\tilde{\epsilon}_{jk}^{(i)}; \mu_{jkm}, \tau_{jkm}), \quad \sum_{m=1}^M \pi_m^{(i)} = 1 \tag{16}$$

Next, for each $j \in [p], k \in [S], m \in [M]$ we update the mean parameter $\mu_{jkm}$ by sampling from a $\mathrm{N}(p_{jkm}, q_{jkm}^{-1})$ distribution with,

$$\begin{aligned} q_{jkm} &= \left(1/b_\mu + \sum_{i=1}^n \mathbf{1}(\xi_{jk}^{(i)} = m)\right) \\ p_{jkm} &= q_{jkm}^{-1}\left(a_\mu + \sum_{i=1}^n \mathbf{1}(\xi_{jk}^{(i)} = m)\tilde{\epsilon}_{jk}^{(i)}\right) \end{aligned} \tag{17}$$

The variance parameter $\tau_{jkm}$ by sampling from a $\mathrm{IG}(p'_{jkm}, q'_{jkm})$ where,

$$\begin{aligned} p'_{jkm} &= a_\tau + 1/2 \sum_{i=1}^n \mathbf{1}(\xi_{jk}^{(i)} = m) \\ q'_{jkm} &= b_\tau + 1/2 \sum_{i=1}^n \mathbf{1}(\xi_{jk}^{(i)} = m)(\tilde{\epsilon}_{jk}^{(i)} - \mu_{jkm})^2 \end{aligned} \tag{18}$$

**Posterior distribution of the orthonormal basis coefficients:** For each $i \in [n]$, define $\boldsymbol{L}^{(i)} = (\boldsymbol{I} - \tilde{\boldsymbol{B}})^\top \boldsymbol{T}^{(i)-1}(\boldsymbol{I} - \tilde{\boldsymbol{B}})$, $\boldsymbol{D}_1^{(i)} = \mathrm{diag}(\boldsymbol{D}_{11}^{(i)}, \ldots, \boldsymbol{D}_{1p}^{(i)})$ with $\boldsymbol{D}_{1j}^{(i)} = \left(\sum_{t\in\mathcal{T}_j^{(i)}} \boldsymbol{\phi}(t)\boldsymbol{\phi}(t)^\top \Big/ \sigma_j\right)$

136 and $\boldsymbol{D}_2^{(i)} = (\boldsymbol{d}_{21}^{(i)\top}, \ldots, \boldsymbol{d}_{2p}^{(i)\top})^\top$ with $\boldsymbol{d}_{2j}^{(i)} = \left( \sum_{t \in \mathcal{T}_j^{(i)}} X_{jt}^{(i)} \phi(t) \Big/ \sigma_j \right)$. Now we sample $\tilde{\boldsymbol{\alpha}}^{(i)}$ from

137 $\mathrm{N}_{pS}(\boldsymbol{p}_\alpha^{(i)}, \boldsymbol{Q}_\alpha^{(i)-1})$ where,

$$
\begin{aligned}
\boldsymbol{Q}_\alpha^{(i)} &= (\boldsymbol{D}_1^{(i)} + \boldsymbol{L}^{(i)}) \\
\boldsymbol{p}_\alpha^{(i)} &= \boldsymbol{Q}_\alpha^{(i)-1} \left( \boldsymbol{D}_2^{(i)} + (\boldsymbol{I} - \tilde{\boldsymbol{B}})' \boldsymbol{T}^{(i)-1} \boldsymbol{M}^{(i)} \right)
\end{aligned}
\tag{19}
$$

138 **Posterior distribution of the noise variances:** For each $j \in [p]$, update $\sigma_j$ by sampling from
139 $\mathrm{IG}(p_\sigma, q_\sigma)$ where,

$$
\begin{aligned}
p_\sigma &= a_\sigma + 1/2 \sum_{i=1}^n T_j^{(i)} \\
q_\sigma &= b_\sigma + 1/2 \sum_{i=1}^n \sum_{t \in \mathcal{T}_j^{(i)}} \left( X_{jt}^{(i)} - \tilde{\boldsymbol{\alpha}}_j^{(i)\top} \phi(t) \right)^2
\end{aligned}
\tag{20}
$$

140 **Posterior distribution of the edge formation probability:** Update the edge probability $r$ by
141 drawing from a $\mathrm{Beta}(p_r, q_r)$ distribution where,

$$
\begin{aligned}
p_r &= a_r + \sum_{j \neq \ell} E_{j\ell} \\
q_r &= b_r + \sum_{j \neq \ell} (1 - E_{j\ell})
\end{aligned}
\tag{21}
$$

142 **Posterior distribution of the causal effect size:** Update $\gamma$ by drawing from a $\mathrm{IG}(p_\gamma, q_\gamma)$ where,

$$
\begin{aligned}
p_\gamma &= a_\gamma + K^2/2 \sum_{j \neq \ell} E_{j\ell} \\
q_\gamma &= b_\gamma + 1/2 \sum_{j \neq \ell} E_{j\ell} \, \mathrm{trace}(\boldsymbol{B}_{j\ell}^\top \boldsymbol{B}_{j\ell})
\end{aligned}
\tag{22}
$$

143 **Posterior distribution of the coefficients of the bspline coefficients:** Define for each $i \in [n], j \in$
144 $[p], k \in [S]$, $\tilde{X}_{jt,-k}^{(i)} = X_{jt}^{(i)} - \sum_{\substack{h=1 \\ h \neq k}}^S \tilde{\alpha}_{jh}^{(i)} \phi_h(t)$. For each $k \in [S]$, we draw $\tilde{\boldsymbol{A}}_k^U$ from $\mathrm{N}_R(\boldsymbol{p}_k, \boldsymbol{Q}_k)$
145 where,

$$
\begin{aligned}
\boldsymbol{Q}_k &= \left[ \left\{ \sum_{i=1}^n \sum_{j=1}^p \frac{(\tilde{\alpha}_{jk}^{(i)})^2}{\sigma_j} \sum_{t \in \mathcal{T}_j^{(i)}} \boldsymbol{b}(t) \boldsymbol{b}(t)^\top \right\} + \boldsymbol{S}_k^{-1} \right]^{-1} \\
\boldsymbol{p}_k &= \boldsymbol{Q}_k \left[ \sum_{i=1}^n \sum_{j=1}^p \sum_{t \in \mathcal{T}_j^{(i)}} \frac{\tilde{\alpha}_{jk}^{(i)}}{\sigma_j} \tilde{X}_{jt,-k}^{(i)} \boldsymbol{b}(t) \right]
\end{aligned}
\tag{23}
$$

146 Now, denote $\boldsymbol{P}_k = \boldsymbol{J} \tilde{\boldsymbol{A}}_{-k}$, where $\boldsymbol{J} = \int \tilde{\boldsymbol{b}}(\omega) \tilde{\boldsymbol{b}}^\top(\omega) \, d\omega$. Finally, transform and normalize the uncon-
147 strained sample to $\tilde{\boldsymbol{A}}_k^N = \tilde{\boldsymbol{A}}_k^U - \boldsymbol{Q}_k \boldsymbol{P}_k (\boldsymbol{P}_k^\top \boldsymbol{Q}_k \boldsymbol{P}_k)^{-1} \boldsymbol{P}_k \tilde{\boldsymbol{A}}_k^U$ and $\tilde{\boldsymbol{A}}_k = \tilde{\boldsymbol{A}}_k^N \times ([\tilde{\boldsymbol{A}}_k^N]^\top \boldsymbol{J} \tilde{\boldsymbol{A}}_k^N)^{-1/2}$.

148 **Posterior distribution of the regularization parameter.** Independently for each $k \in [S]$, con-
149 ditional on all other parameters, denote $q_k = 1/2 \sum_{r=3}^R \tilde{A}_{kr}^2$ and $p = R/2$. We then draw each $\lambda_k$
150 from a $\mathrm{Gamma}(p, q_k)$ distribution truncated at $(L_k, U_k)$.

**Posterior distribution of the adjacency and causal effect matrices.** Recursively for each $E_{j\ell}, j, \ell \in [p]$, we perform a birth/death move such that $\boldsymbol{E}' = \boldsymbol{E}$ except $E'_{j\ell} = 1 - E_{j\ell}$. The joint posterior of $(\boldsymbol{B}_{j\ell}, E_{j\ell})$ does not have closed form expression and therefore we perform a Metropolis Hastings (MH) step for joint acceptance or rejection of $(\boldsymbol{B}_{j\ell}, E_{j\ell})$. First we draw a $\boldsymbol{B}'_{j\ell}$ from a proposal distribution $N(\boldsymbol{B}_{j\ell}, z\boldsymbol{I}_K, \boldsymbol{I}_K)$. We check whether $\boldsymbol{B}'(= \boldsymbol{B}$ except for $j, \ell$ block entry) satisfies the eigenvalue condition given in Assumption 2 of the main manuscript. If yes then we proceed to the next step and if not, we draw another $\boldsymbol{B}'_{j\ell}$ from the proposal ditribution. Here $z$ is a tuning parameter for the MH step. Next we calculate the acceptance ratio$(\alpha) = \alpha_N - \alpha_D$ where,

$$\alpha_N = E'_{j\ell} \log\left(r MVN(\boldsymbol{B}'_{j\ell}; \boldsymbol{B}_{j\ell}, \gamma\boldsymbol{I}_K, \boldsymbol{I}_K)\right) + (1-E'_{j\ell}) \log\left((1-r)MVN(\boldsymbol{B}'_{j\ell}; \boldsymbol{B}_{j\ell}, s\gamma\boldsymbol{I}_K, \boldsymbol{I}_K)\right) +$$
$$\sum_{i=1}^{n} \log\left(N(\tilde{\boldsymbol{\alpha}}^{(i)}; (\boldsymbol{I} - \tilde{\boldsymbol{B}}')^{-1}\boldsymbol{M}^{(i)}, (\boldsymbol{I} - \tilde{\boldsymbol{B}}')^{\top}\boldsymbol{T}^{(i)-1}(\boldsymbol{I} - \tilde{\boldsymbol{B}}'))\right) \quad (24)$$

$$\alpha_D = E_{j\ell} \log\left(r MVN(\boldsymbol{B}_{j\ell}; \boldsymbol{B}'_{j\ell}, \gamma\boldsymbol{I}_K, \boldsymbol{I}_K)\right) + (1-E_{j\ell}) \log\left((1-r)MVN(\boldsymbol{B}_{j\ell}; \boldsymbol{B}'_{j\ell}, s\gamma\boldsymbol{I}_K, \boldsymbol{I}_K)\right) +$$
$$\sum_{i=1}^{n} \log\left(N(\tilde{\boldsymbol{\alpha}}^{(i)}; (\boldsymbol{I} - \tilde{\boldsymbol{B}})^{-1}\boldsymbol{M}^{(i)}, (\boldsymbol{I} - \tilde{\boldsymbol{B}})^{\top}\boldsymbol{T}^{(i)-1}(\boldsymbol{I} - \tilde{\boldsymbol{B}}))\right) \quad (25)$$

Then we accept or reject the proposed $(\boldsymbol{B}'_{j\ell}, E'_{j\ell})$ based on whether the value of a uniform random variable is less than or greater than $\min\{1, \alpha\}$. The value of $z$ is tuned to achieve an acceptance rate between 20% to 40%.

## B.3 Markov Chain Monte Carlo algorithm

In this section we delineate the steps of Markov Chain Monte Carlo algorithm for drawing samples from the posterior distributions.

---

**Algorithm 1** MCMC algorithm to obtain posterior samples

1: **for** $b \leftarrow 1$ to $B$ **do**
2:     **for** $i \leftarrow 1$ to $n$ **do**
3:         Draw $\tilde{\boldsymbol{\alpha}}^{(i),[b]} \sim \mathrm{N}_{pS}(\boldsymbol{p}_\alpha^{(i)}, (\boldsymbol{Q}_\alpha^{(i)})^{-1})$;         ▷ Update the basis coefficients by (19)
4:     **end for**
5: **end for**
6: **for** $j \leftarrow 1$ to $p$ **do**
7:     Draw $\sigma_j^{[b]}$ from IG $(p_\sigma, q_\sigma)$;             ▷ Update the noise variances by (20)
8: **end for**
9: Draw $r^{[b]}$ from Beta$(p_r, q_r)$;         ▷ Update the edge formation probability by (21)
10: Draw $\gamma^{[b]}$ from IG$(p_\gamma, q_\gamma)$;         ▷ Update the causal effect size by (22)
11: **for** $k \leftarrow 1$ to $S$ **do**
12:     Draw $\tilde{\boldsymbol{A}}_k^U \sim \mathrm{N}_R(\boldsymbol{p}_k, \boldsymbol{Q}_k)$;    ▷ Update the un-normalized bspline coefficients by (23)
13:     Calculate $\boldsymbol{P}_k = \boldsymbol{J}\tilde{\boldsymbol{A}}_{-k}$;
14:     Calculate $\tilde{\boldsymbol{A}}_k^N = \tilde{\boldsymbol{A}}_k^U - \boldsymbol{Q}_k\boldsymbol{P}_k(\boldsymbol{P}_k^{\top}\boldsymbol{Q}_k\boldsymbol{P}_k)^{-1}\boldsymbol{P}_k\tilde{\boldsymbol{A}}_k^U$;
15:     Normalize $\tilde{\boldsymbol{A}}_k^{[b]} = \tilde{\boldsymbol{A}}_k^N \times ([\tilde{\boldsymbol{A}}_k^N]^{\top}\boldsymbol{J}\tilde{\boldsymbol{A}}_k^N)^{-1/2}$;
16: **end for**
17: **for** $k \leftarrow 1$ to $S$ **do**
18:     Draw $\lambda_k^{[b]} \sim \mathrm{Gamma}\left(\frac{R}{2}, \frac{\sum_{r=3}^{R}\left(\tilde{A}_{kr}^{[b]}\right)^2}{2}\right)$;       ▷ Update the regularization parameter
19: **end for**

---

```
20: for j ← 1 to p do
21:     for k ← 1 to K do
22:         Draw π_{jk}^{[b]} ~ Dir(β_1, ..., β_M)                    ▷ Update the mixing weights by (15)
23:         for i ← 1 to n do
24:             Draw ξ_{jk}^{(i),[b]} ~ Cat({π_m^{(i),[b]}}_{m∈[M]});    ▷ Update the class labels by (16)
25:         end for
26:     end for
27: end for
28: for j ← 1 to p do
29:     for k ← 1 to K do
30:         for m ← 1 to M do
31:             Draw μ_{jkm}^{[b]} ~ N(p_{jkm}, q_{jkm}^{-1});          ▷ Update the mean parameter by (17)
32:             Draw τ_{jkm}^{[b]} ~ IG(p'_{jkm}, q'_{jkm});           ▷ Update the variance parameter by (18)
33:         end for
34:     end for
35: end for
36: for j ← 1 to p do
37:     for ℓ ← 1 to p do
38:         Update (E_{jℓ}, B_{jℓ}) by MH step using (25) and (24)
39:     end for
40: end for
```

## C  Some additional simulations

### C.1  Misspecification analysis of the proposed model

#### C.1.1  With general exogenous variable distributions

In this section we consider simulating the exogenous variable from distributions other than that of laplace distribution and compare the performance of our algorithm. In particular, following Shimizu et al., 2011, we generate the exogenous variable $\epsilon_{jk}$ from (1) Student t distribution with 1 degrees of freedom, (2) Uniform (3) Exponential, (4) Mixture of two double exponentials, (5) Symmetric mixture of four Gaussians, and (6) Non symmetric mixture of two Gaussians. Across all exogenous variable distributions, Table 1 shows that the proposed FENCE model had the best performance.

Table 1: Table showing comparison of several methods for different distributions of exogenous variables $\epsilon_{jk}$ under 50 replicates

| Distributions | FENCE | | | fLiNG | | | fPCA-LINGAM | | | fPCA-PC | | | fPCA-CCD | | |
|---|---|---|---|---|---|---|---|---|---|---|---|---|---|---|---|
| | TPR | FDR | MCC | TPR | FDR | MCC | TPR | FDR | MCC | TPR | FDR | MCC | TPR | FDR | MCC |
| (1) | 0.81(0.04) | 0.24(0.07) | 0.76(0.05) | 0.71(0.09) | 0.69(0.05) | 0.36(0.04) | 0.85(0.02) | 0.84(0.04) | 0.28(0.08) | 0.81(0.05) | 0.71(0.06) | 0.26(0.07) | 0.91(0.02) | 0.62(0.04) | 0.38(0.02) |
| (2) | 0.75(0.04) | 0.21(0.03) | 0.86(0.04) | 0.73(0.04) | 0.68(0.04) | 0.33(0.07) | 0.82(0.06) | 0.76(0.04) | 0.26(0.02) | 0.83(0.06) | 0.67(0.04) | 0.30(0.05) | 0.87(0.02) | 0.69(0.05) | 0.35(0.04) |
| (3) | 0.77(0.04) | 0.23(0.05) | 0.83(0.04) | 0.74(0.04) | 0.63(0.03) | 0.32(0.04) | 0.86(0.04) | 0.81(0.03) | 0.24(0.03) | 0.81(0.03) | 0.76(0.03) | 0.31(0.03) | 0.89(0.02) | 0.73(0.05) | 0.41(0.03) |
| (4) | 0.88(0.07) | 0.14(0.06) | 0.89(0.05) | 0.67(0.07) | 0.75(0.06) | 0.29(0.05) | 0.81(0.02) | 0.79(0.05) | 0.22(0.09) | 0.82(0.08) | 0.75(0.04) | 0.27(0.05) | 0.83(0.03) | 0.58(0.03) | 0.43(0.03) |
| (5) | 0.81(0.07) | 0.21(0.06) | 0.87(0.05) | 0.69(0.06) | 0.71(0.05) | 0.25(0.04) | 0.84(0.03) | 0.76(0.02) | 0.25(0.06) | 0.80(0.07) | 0.73(0.05) | 0.29(0.03) | 0.86(0.04) | 0.67(0.05) | 0.36(0.02) |
| (6) | 0.79(0.06) | 0.24(0.05) | 0.81(0.04) | 0.70(0.04) | 0.71(0.06) | 0.28(0.05) | 0.82(0.04) | 0.73(0.05) | 0.31(0.03) | 0.83(0.07) | 0.71(0.05) | 0.25(0.03) | 0.78(0.02) | 0.68(0.04) | 0.39(0.05) |

### C.1.2  With functions observed on unevenly spaced grids

In this experiment, we generated simulated data with $(n, p)$ values of either $(500, 20)$, $(500, 50)$, $(800, 20)$, or $(800, 50)$. Unlike the method used in Section 4 of the main manuscript, we initially selected 250 points at random from the uniform distribution between 0 and 1 and defined this set as $D$. For each realization $i$ of function $j$, we randomly selected a subset $D_j^{(i)}$ of size $m_j^{(i)} = 20$ from $D$ to measure the function. We generated the causal graph, direct causal effect matrix, orthonormal basis functions, basis coefficient sequences, and observations in the same way as Section 4 of the main manuscript. We conducted this scenario 50 times and compared the results with those from fLiNG, fPCA-LiNGAM, fPCA-PC and fPCA-CCD. The results presented in Table 2 demonstrate that FENCE is effective and superior to these other methods in learning directed cyclic graphs for general multivariate functional data.

Table 2: Comaprison of various methods under unevenly specified grids under 50 replicates

| n | p | FENCE | | | fLiNG | | | fPCA-LINGAM | | | fPCA-PC | | | fPCA-CCD | | |
|---|---|---|---|---|---|---|---|---|---|---|---|---|---|---|---|---|
| | | TPR | FDR | MCC | TPR | FDR | MCC | TPR | FDR | MCC | TPR | FDR | MCC | TPR | FDR | MCC |
| 500 | 20 | 0.86(0.03) | 0.19(0.05) | 0.89(0.05) | 0.46(0.09) | 0.73(0.05) | 0.32(0.03) | 0.25(0.02) | 0.82(0.04) | 0.17(0.04) | 0.21(0.04) | 0.83(0.04) | 0.19(0.07) | 0.56(0.02) | 0.62(0.05) | 0.32(0.06) |
| 500 | 50 | 0.79(0.04) | 0.24(0.06) | 0.84(0.04) | 0.37(0.04) | 0.79(0.06) | 0.29(0.05) | 0.23(0.04) | 0.87(0.03) | 0.15(0.02) | 0.17(0.06) | 0.85(0.04) | 0.17(0.05) | 0.51(0.05) | 0.67(0.03) | 0.27(0.01) |
| 800 | 20 | 0.91(0.04) | 0.16(0.03) | 0.91(0.04) | 0.61(0.07) | 0.79(0.06) | 0.41(0.04) | 0.33(0.03) | 0.79(0.05) | 0.25(0.02) | 0.35(0.03) | 0.81(0.02) | 0.31(0.03) | 0.78(0.02) | 0.56(0.01) | 0.39(0.03) |
| 800 | 50 | 0.88(0.07) | 0.20(0.04) | 0.88(0.05) | 0.55(0.03) | 0.81(0.02) | 0.38(0.05) | 0.27(0.02) | 0.86(0.05) | 0.22(0.09) | 0.31(0.06) | 0.82(0.04) | 0.29(0.05) | 0.73(0.03) | 0.64(0.04) | 0.36(0.05) |

### C.1.3 When the true graph is acyclic

In this section we compared our method with the fLiNG method under the assumption that the true graph is acyclic. The entire simulation setting remains same as that of described in Section 4 of the main manuscript except that the true graph was generated under acyclicity constraint. It is observed from Table 3 that under this assumption, fLiNG has superior performance against the proposed FENCE model.

Table 3: Comparison of two methods when the true graph is acyclic under 50 replicates

| n | p | d | FENCE | | | fLiNG | | |
|---|---|---|---|---|---|---|---|---|
| | | | TPR | FDR | MCC | TPR | FDR | MCC |
| 150 | 30 | 125 | 0.81(0.03) | 0.29(0.05) | 0.85(0.05) | 0.83(0.05) | 0.21(0.05) | 0.91(0.04) |
| 150 | 60 | 125 | 0.79(0.04) | 0.32(0.06) | 0.81(0.02) | 0.82(0.03) | 0.24(0.06) | 0.87(0.05) |
| 150 | 30 | 250 | 0.67(0.05) | 0.34(0.06) | 0.79(0.04) | 0.81(0.04) | 0.23(0.06) | 0.82(0.05) |
| 150 | 60 | 250 | 0.64(0.04) | 0.36(0.03) | 0.74(0.04) | 0.78(0.05) | 0.26(0.06) | 0.79(0.05) |
| 300 | 30 | 125 | 0.85(0.03) | 0.23(0.05) | 0.87(0.04) | 0.79(0.05) | 0.19(0.06) | 0.93(0.04) |
| 300 | 60 | 125 | 0.81(0.04) | 0.26(0.05) | 0.81(0.08) | 0.85(0.02) | 0.21(0.05) | 0.87(0.04) |
| 300 | 30 | 250 | 0.77(0.02) | 0.31(0.06) | 0.81(0.05) | 0.78(0.03) | 0.23(0.06) | 0.85(0.05) |
| 300 | 60 | 250 | 0.75(0.03) | 0.35(0.04) | 0.79(0.05) | 0.82(0.03) | 0.24(0.05) | 0.80(0.05) |

### C.1.4 When the true structural equation model is non-linear

In this section, we have outlined the misspecification analysis for our model by generating data corresponding to a non-linear structural equation model (SEM). We have considered a scenario with number of samples $(n) = 100$, number of nodes $(p) = 6$ and evenly spaced time grid $(d)$ over $(0, 1)$ of size $d = 100$. The summary measures corresponding to 10 replicates are given in Table 4 below. The poor performance is clearly expected because our modeling assumptions involve linear SEM.

Table 4: Performance of FENCE when the true SEM is non-linear

| FENCE | | |
|---|---|---|
| TPR | FDR | MCC |
| 0.27(0.08) | 0.71(0.07) | 0.34(0.07) |

### C.2 Sensitivity analysis

In this section, we outline how sensitive the performance of our model is against different choices of hyperparameters. The hyperparameters for our model are $(a_\gamma, b_\gamma), (a_\tau, b_\tau), (a_\sigma, b_\sigma), s, R, S, M$ and $\beta$. The data were generated the same way as in Section 4 of the main manuscript with $(n, p, d) = (150, 20, 125)$. From Table 5 we can conclude that the performance of our model is quite robust under different choice of hyperparameters.

## D Comparison of various methods

In this section, as discussed in Section 4 of the main manuscript, we give the full summary of the simulation results related to the comparison of our method, FENCE, against fLiNG, fPCA-LiNGAM, fPCA-PC and fPCA-CCD in Table 6. Our conclusions remain the same.

Table 5: Sensitivity analysis for different choices of hyperparameters. The metrics reported are based on 50 repetitions are reported; standard deviations are given within the parentheses.

| Hyperparameters | $(a_\tau, b_\tau) = (0.1, 0.1)$ | $(a_\sigma, b_\sigma) = (0.1, 0.1)$ | $(a_\gamma, b_\gamma) = (0.1, 0.1)$ | $s = 0.01$ | $R = 30$ | $S = 15$ | $M = 15$ | $\beta = 0.1$ |
|---|---|---|---|---|---|---|---|---|
| TPR | 0.79(0.02) | 0.80(0.02) | 0.78(0.03) | 0.75(0.03) | 0.79(0.02) | 0.80(0.01) | 0.81(0.02) | 0.82(0.02) |
| FDR | 0.16(0.03) | 0.18(0.03) | 0.18(0.05) | 0.20(0.04) | 0.23(0.02) | 0.19(0.03) | 0.15(0.03) | 0.21(0.02) |
| MCC | 0.76(0.04) | 0.81(0.04) | 0.83(0.04) | 0.82(0.03) | 0.81(0.01) | 0.84(0.04) | 0.83(0.01) | 0.84(0.03) |
| Hyperparameters | $(a_\tau, b_\tau) = (0.1, 1)$ | $(a_\sigma, b_\sigma) = (0.01, 0.01)$ | $(a_\gamma, b_\gamma) = (0.1, 1)$ | $s = 0.03$ | $R = 20$ | $S = 10$ | $M = 20$ | $\beta = 2$ |
| TPR | 0.80(0.02) | 0.79(0.03) | 0.76(0.02) | 0.75(0.03) | 0.78(0.03) | 0.79(0.02) | 0.82(0.04) | 0.81(0.03) |
| FDR | 0.17(0.03) | 0.18(0.04) | 0.15(0.03) | 0.19(0.04) | 0.22(0.03) | 0.21(0.03) | 0.15(0.03) | 0.23(0.03) |
| MCC | 0.77(0.02) | 0.80(0.02) | 0.82(0.02) | 0.82(0.03) | 0.80(0.02) | 0.83(0.03) | 0.83(0.03) | 0.85(0.05) |
| Hyperparameters | $(a_\tau, b_\tau) = (5, 1)$ | $(a_\sigma, b_\sigma) = (0.1, 1)$ | $(a_\gamma, b_\gamma) = (5, 1)$ | $s = 0.05$ | $R = 25$ | $S = 20$ | $M = 30$ | $\beta = 5$ |
| TPR | 0.76(0.04) | 0.82(0.05) | 0.79(0.03) | 0.76(0.04) | 0.78(0.03) | 0.80(0.03) | 0.81(0.02) | 0.79(0.04) |
| FDR | 0.17(0.05) | 0.19(0.04) | 0.19(0.02) | 0.20(0.04) | 0.23(0.03) | 0.22(0.04) | 0.16(0.03) | 0.21(0.03) |
| MCC | 0.78(0.02 ) | 0.83(0.03) | 0.83(0.05) | 0.81(0.03) | 0.81(0.01) | 0.84(0.01) | 0.82(0.02) | 0.83(0.03) |

Table 6: Comparison of performance of various methods under 50 replicates. Since LiNGAM is not applicable to cases where $q > n$ with $q = Kp$ being the total number of extracted basis coefficients across all functions, the results from those cases are not available and indicated by "-".

| n | p | d | FENCE | | | fLiNG | | | fPCA-LINGAM | | | fPCA-PC | | | fPCA-CCD | | |
|---|---|---|---|---|---|---|---|---|---|---|---|---|---|---|---|---|---|
| | | | TPR | FDR | MCC | TPR | FDR | MCC | TPR | FDR | MCC | TPR | FDR | MCC | TPR | FDR | MCC |
| 75 | 20 | 125 | **0.85(0.09)** | **0.19(0.07)** | **0.88(0.05)** | 0.41(0.09) | 0.79(0.05) | 0.36(0.04) | 0.35(0.19) | 0.84(0.04) | 0.11(0.08) | 0.20(0.09) | 0.83(0.06) | 0.10(0.07) | 0.69(0.03) | 0.41(0.04) | 0.23(0.03) |
| 75 | 40 | 125 | **0.79(0.08)** | **0.23(0.06)** | **0.86(0.04)** | 0.37(0.08) | 0.82(0.06) | 0.33(0.05) | - | - | - | 0.11(0.06) | 0.91(0.04) | 0.05(0.05) | 0.73(0.02) | 0.47(0.04) | 0.21(0.05) |
| 75 | 60 | 125 | **0.75(0.07)** | **0.27(0.05)** | **0.83(0.04)** | 0.34(0.07) | 0.83(0.06) | 0.32(0.04) | - | - | - | 0.11(0.03) | 0.91(0.03) | 0.06(0.03) | 0.68(0.03) | 0.61(0.05) | 0.19(0.03) |
| 150 | 20 | 125 | **0.88(0.07)** | **0.14(0.06)** | **0.89(0.05)** | 0.45(0.07) | 0.75(0.06) | 0.39(0.05) | 0.28(0.22) | 0.86(0.05) | 0.08(0.09) | 0.31(0.08) | 0.75(0.04) | 0.12(0.05) | 0.71(0.03) | 0.42(0.03) | 0.25(0.04) |
| 150 | 40 | 125 | **0.81(0.07)** | **0.21(0.06)** | **0.87(0.05)** | 0.39(0.06) | 0.79(0.05) | 0.37(0.04) | 0.35(0.22) | 0.91(0.02) | 0.08(0.06) | 0.25(0.07) | 0.81(0.05) | 0.06(0.03) | 0.73(0.04) | 0.47(0.05) | 0.23(0.03) |
| 150 | 60 | 125 | **0.79(0.06)** | **0.24(0.05)** | **0.86(0.04)** | 0.36(0.04) | 0.80(0.06) | 0.36(0.05) | - | - | - | 0.23(0.07) | 0.83(0.05) | 0.05(0.03) | 0.72(0.05) | 0.54(0.04) | 0.22(0.02) |
| 300 | 20 | 125 | **0.91(0.03)** | **0.09(0.04)** | **0.90(0.04)** | 0.51(0.04) | 0.73(0.06) | 0.41(0.04) | 0.30(0.19) | 0.84(0.05) | 0.11(0.09) | 0.36(0.09) | 0.72(0.05) | 0.14(0.05) | 0.81(0.03) | 0.39(0.04) | 0.26(0.04) |
| 300 | 40 | 125 | **0.87(0.04)** | **0.15(0.05)** | **0.87(0.05)** | 0.47(0.05) | 0.75(0.06) | 0.38(0.05) | 0.27(0.20) | 0.91(0.02) | 0.08(0.06) | 0.29(0.06) | 0.76(0.06) | 0.07(0.03) | 0.77(0.03) | 0.45(0.02) | 0.24(0.03) |
| 300 | 60 | 125 | **0.85(0.05)** | **0.17(0.03)** | **0.86(0.03)** | 0.45(0.05) | 0.76(0.04) | 0.38(0.03) | 0.28(0.17) | 0.91(0.05) | 0.07(0.06) | 0.28(0.04) | 0.77(0.05) | 0.05(0.03) | 0.72(0.03) | 0.49(0.02) | 0.22(0.03) |
| 75 | 20 | 250 | **0.81(0.04)** | **0.23(0.02)** | **0.85(0.05)** | 0.39(0.07) | 0.80(0.05) | 0.39(0.04) | 0.32(0.14) | 0.82(0.03) | 0.09(0.04) | 0.19(0.07) | 0.81(0.04) | 0.13(0.07) | 0.67(0.03) | 0.46(0.03) | 0.22(0.04) |
| 75 | 40 | 250 | **0.73(0.04)** | **0.28(0.05)** | **0.82(0.04)** | 0.35(0.04) | 0.85(0.06) | 0.33(0.05) | - | - | - | 0.25(0.06) | 0.83(0.04) | 0.12(0.04) | 0.68(0.02) | 0.51(0.04) | 0.21(0.04) |
| 75 | 60 | 250 | **0.67(0.03)** | **0.34(0.05)** | **0.79(0.04)** | 0.34(0.04) | 0.85(0.03) | 0.31(0.04) | - | - | - | 0.17(0.03) | 0.83(0.02) | 0.09(0.03) | 0.63(0.04) | 0.56(0.04) | 0.19(0.04) |
| 150 | 20 | 250 | **0.83(0.06)** | **0.17(0.05)** | **0.86(0.05)** | 0.46(0.07) | 0.73(0.07) | 0.42(0.05) | 0.32(0.19) | 0.79(0.05) | 0.13(0.05) | 0.41(0.08) | 0.72(0.04) | 0.19(0.05) | 0.73(0.04) | 0.43(0.02) | 0.24(0.03) |
| 150 | 40 | 250 | **0.79(0.02)** | **0.26(0.06)** | **0.82(0.03)** | 0.41(0.05) | 0.71(0.05) | 0.40(0.03) | 0.31(0.14) | 0.81(0.02) | 0.13(0.06) | 0.46(0.07) | 0.73(0.05) | 0.15(0.02) | 0.71(0.03) | 0.47(0.04) | 0.23(0.03) |
| 150 | 60 | 250 | **0.69(0.05)** | **0.31(0.05)** | **0.79(0.04)** | 0.43(0.03) | 0.79(0.06) | 0.43(0.05) | - | - | - | 0.41(0.03) | 0.75(0.05) | 0.14(0.03) | 0.69(0.02) | 0.52(0.03) | 0.21(0.02) |
| 300 | 20 | 250 | **0.86(0.02)** | **0.16(0.04)** | **0.85(0.04)** | 0.68(0.02) | 0.77(0.07) | 0.47(0.04) | 0.45(0.13) | 0.86(0.05) | 0.17(0.09) | 0.42(0.09) | 0.86(0.03) | 0.13(0.05) | 0.78(0.02) | 0.44(0.06) | 0.27(0.03) |
| 300 | 40 | 250 | **0.79(0.08)** | **0.16(0.05)** | **0.84(0.06)** | 0.73(0.05) | 0.71(0.06) | 0.43(0.05) | 0.39(0.16) | 0.87(0.05) | 0.16(0.07) | 0.45(0.06) | 0.81(0.06) | 0.12(0.06) | 0.76(0.05) | 0.49(0.06) | 0.23(0.05) |
| 300 | 60 | 250 | 0.76(0.05) | **0.21(0.03)** | **0.80(0.03)** | **0.77(0.05)** | 0.74(0.03) | 0.42(0.03) | 0.28(0.17) | 0.90(0.04) | 0.13(0.04) | 0.43(0.04) | 0.79(0.07) | 0.12(0.04) | 0.72(0.06) | 0.53(0.03) | 0.22(0.04) |