# OpenReview forum: "Directed Cyclic Graph for Causal Discovery from Multivariate Functional Data"
_NeurIPS.cc/2023/Conference — NeurIPS 2023 poster_

### Official Review · Reviewer_Y9Hq · 2023-06-19

**Soundness:** 3 good
**Presentation:** 1 poor
**Contribution:** 2 fair
**Rating:** 5
**Confidence:** 3

**Summary:**

The paper develops a causal discovery method for directed cyclic graph. The proposed method is a two-step approach which utilizes a Bayesian approach to reduce the dimension of functional data and performs the causal structure learning on the learned embeddings. Experiments on the synthetic and real-world datasets well demonstrate the effectiveness of the proposed method.

**Strengths:**

1. There are seldom paper studying the causal discovery method on the directed cyclic graph. However, in some particular applications, there may exist cycles in the causal graphs.
2. The proposed method outperforms baselines in various settings.
3. The results of the brain EEG data provide some insights on the organization of brain regions.


**Weaknesses:**

1. It lacks some classical and SOTA baselines, e.g., NOTEARS[1], DAG-GNN[2], CSIvA[3]
[1] Zheng X, Aragam B, Ravikumar P K, et al. Dags with no tears: Continuous optimization for structure learning[J]. Advances in neural information processing systems, 2018, 31.
[2] Yu Y, Chen J, Gao T, et al. DAG-GNN: DAG structure learning with graph neural networks[C]//International Conference on Machine Learning. PMLR, 2019: 7154-7163.
[3] Learning to induce causal structure. ICLR 2023.
2. It is hard to set so many priors for the proposed model in practice.
3. The presentation of the manuscript could be improved. Fig.1 is hard to understand.
4. For the results of real data, it is better to also show the results of baselines.


**Questions:**

Please refer to weaknesses.

**Limitations:**

No
I have not found the limitations and  broader societal impacts of this manuscript.

---

> ### Author Rebuttal · Authors · 2023-08-09
>
> **[Q1]**  Following your suggestions, we have now compared the proposed method FENCE with DAG-GNN and NOTEARS (we could not find publicly available code for CSIvA yet). Both DAG-GNN and NOTEARS were applied to the functional principle components. The results are reported in the table below. We have considered the scenario where the number of samples $n = 150$, number of nodes $p = 40$, and evenly spaced time grid over $(0, 1)$ of size $d = 125$. These results are based on $10$ replicates.
>
> | Method | TPR | FDR | MCC |
> | --------- | :-----: | :-----: | :------: |
> | FENCE | 0.80(0.07) | 0.23(0.06) | 0.83(0.05) |
> | DAG-GNN | 0.34(0.03) | 0.63(0.07) | 0.41(0.08) |
> | NOTEARS | 0.38(0.05) | 0.58(0.05) | 0.47(0.09) |
>
> We can see that FENCE has outperformed both DAG-GNN and NOTEARS for the above considered scenario, again indicating the ineffectiveness of approaches that use functional PCA + existing multivariate causal discovery methods.
>
> **[Q2]** The default choice for hyperparameters that we took are:
>
> * Causal effect size $\gamma$: $a\_{\gamma} = 1, b\_{\gamma} = 1$
> * Noise variance $\sigma\_{j}$: $a\_{\sigma} = 0.01, b\_{\sigma} = 0.01$
> * Variance of the Gaussian mixture distribution $\tau\_{jkm}$: $a\_{\tau} = 1, b\_{\tau} = 1$
> * Number of known adaptive spline bases $R$: $R = 15$
> * Number of unknown basis functions $S$: $S  = 10$
> * Number of mixture components $M$: $M = 10$
> * Mixing coefficient parameter $\pi\_{jk}$: $\beta = 1$
>
> Many of those choices are standard in Bayesian statistics, for example, $a\_{\sigma} = 0.01, b\_{\sigma} = 0.01$ ensures non-informativeness of the prior on noise variance (i.e., the data will dominate the inference of the noise variance). We have observed that they work well in all of our numerical experiments. From our experience, our method is not highly sensitive to the tuning of these hyperparameters. In Section C.4 of the supplementary materials, we have conducted a sensitivity analysis of those hyperparameters. It shows that our method is relatively robust to hyperparameter choices within the tested ranges. The various options of the hyperparameters that we considered in the sensitivity analysis are listed as follows:
>
> * Causal effect size $\gamma$: $(a\_{\gamma}, b\_{\gamma}) \in \{(0.1, 0.1), (0.1, 1), (5, 1)\}$
> * Noise variance $\sigma\_{j}$: $(a\_{\sigma}, b\_{\sigma}) \in \{(0.1, 0.1), (0.01, 0.01), (0.1, 1)\}$
> * Variance of the Gaussian mixture distribution $\tau\_{jkm}$: $(a\_{\tau}, b\_{\tau}) \in \{(0.1, 0.1), (0.1, 1), (5, 1)\}$
> * Number of known adaptive spline bases $R$: $R \in \{20, 25, 30\}$
> * Number of unknown basis functions $S$: $S \in \{10, 15, 20\}$
> * Number of mixture components $M$: $M \in \{15, 20, 30\}$
> * Mixing coefficient parameter $\pi\_{jk}$: $\beta \in \{0.1, 2, 5\}$
>
> We hope this sensitivity analysis would alleviate your concerns.
>
> **[Q3]** Figure 1 was meant to illustrate the two crucial steps in establishing Theorem 2.1, which we hope helps readers understand the high-level idea of our proof. On the left-hand side (LHS) of the diagram, we depict the hypergraph-like structure that emerges when assuming the existence of disjoint cycles, whereas, on the right-hand side (RHS), we offer a magnified view of the hypernodes (nodes containing simple cycles). Our approach to proving causal identifiability progresses from the LHS to the RHS. That is, we first prove the identifiability of the hypergraph-like structure depicted on the LHS of Figure 1, and then we proceed to establish the identifiability of each simple cycle within every hypernode in the hypergraph. We will add these explanations in the caption of Figure 1 in the revision of our paper.
>
> **[Q4]** We have added a pdf containing the results of the real data analysis for the baselines, together with our response. We will add these results to the revision of our paper.

---

> > ### Comment · Reviewer_Y9Hq · 2023-08-14
> >
> > The authors have properly solved my concerns. Hence I raise my score accordingly.

---

> > > ### Author Response · Authors · 2023-08-15
> > >
> > > Thank you for your reply. We are glad that our response helps. We will revise our work accordingly.

---

### Official Review · Reviewer_3JSM · 2023-06-28

**Soundness:** 3 good
**Presentation:** 3 good
**Contribution:** 3 good
**Rating:** 6
**Confidence:** 3

**Summary:**

This paper discusses the inference of causal relationships between multivariate functions through the proposal of a causal discovery model. The model involves embedding the functional nodes into a lower-dimensional space and separating the structural equation model into two components: the projection onto the space and its orthogonal complement in the Hilbert space defined on the domain. The identifiability of the proposed model is proven based on several assumptions, including disjoint cycles, as described in the Supplementary Materials. The paper presents a model inference method utilizing a fully Bayesian approach. Experimental results using simulated data demonstrate that the proposed method outperforms both conventional causal discovery methods for multivariable functional data and other conventional causal discovery methods. Additionally, insightful observations are made from the application of the proposed method to brain EEG data.

**Strengths:**

The strongest aspect of this paper is its introduction of a causal discovery model for multivariate functional data, which accommodates the presence of cycles. This is a significant contribution since many multivariate functional datasets, such as EEG data, inherently involve cycles, including feedback loops. Additionally, the paper proves the identifiability of the model and presents a fully Bayesian approach to model inference. The effectiveness of the proposed method is demonstrated through the utilization of both simulated data and real-world EEG data.

**Weaknesses:**

The discussion of the results of the brain EEG data is inadequate. While there are differences in connectivity between the alcoholic and control groups, readers are unable to determine the significance of these findings because most readers may not possess a background in brain science.

There may be a minor mistake: from my understanding, the subscript $j=1$ below the superscript $m_j$ in line 116 might actually be $u=1$.

**Questions:**

Is it possible to consider providing additional discussion on the results of the brain EEG data, in order to help readers ascertain the validity of the findings and comprehend their significance?

**Limitations:**

The limitation of the proposed method lies in the assumptions required to establish the identifiability of the model.

---

> ### Author Rebuttal · Authors · 2023-08-09
>
> **Response to your question**:
>
> In Hayden et al. (2006), it was observed that alcohol-dependent subjects exhibited frontal asymmetry, distinguishing them from the control group. Our own investigation aligns well with these results, as we have identified denser connectivity across various brain regions in the middle and left areas of the frontal lobe among alcoholic subjects when compared to controls. Furthermore, Winterer et al. (2003) documented coherent differences between alcoholics and controls in the posterior hemispheres, specifically in the temporal, parietal, and occipital lobes. In accordance with their findings, our study provides additional support for this claim, as we have observed heightened activity with several cycles formed in those same regions within the alcoholic group when compared to the control group.
>
> * Hayden, Elizabeth P., et al. "Patterns of regional brain activity in alcohol‐dependent subjects." Alcoholism: Clinical and Experimental Research 30.12 (2006): 1986-1991.
>
> * Winterer, G., et al. "EEG phenotype in alcoholism: increased coherence in the depressive subtype." Acta Psychiatrica Scandinavica 108.1 (2003): 51-60.
>
> **Response to the weaknesses mentioned**:
>
> We have provided additional validity of our results in the EEG data, as stated in our response above. In addition, we will correct the typo (in line 116) in the revision of our paper.

---

> > ### Comment · Reviewer_3JSM · 2023-08-21
> > **Thank you for your rebuttal!**
> >
> > Thank you very much for your answer to my question! I understood the validity of the experimental results on the brain EEG data. I re-think my opinion by considering the reviews of other reviewers, I decide to change my rating. I sincerely apologize for my late response.

---

### Official Review · Reviewer_nZkc · 2023-07-05

**Soundness:** 3 good
**Presentation:** 3 good
**Contribution:** 3 good
**Rating:** 7
**Confidence:** 3

**Summary:**

The authors propose a causal model, i.e. a linear structural equation model for multivariate functional data. The authors' proposed model does not require the dependency structure to conform to a DAG, allowing for modeling cyclic cause-effect relationships. Given certain assumptions on exogenous variables and cycle structure, the authors show the identifiability of their model; a central assumption of which is that the causal interrelations between the variables pertain only to a subset of the function space which the observations inhabit. The authors propose an MCMC sampling procedure to sample from the posterior of the parameters given prior hyperparameters. The authors test their model under various conditions in synthetic data, in addition to a real world application.

**Strengths:**

- The authors' research is soundly motivated, the central observation regarding the importance of being able to handle cyclic relationships in functional data is well-justified, especially given the application areas they consider.
- The authors' central modeling assumption, that the causal interrelations pertain to a subset of the function space is interesting and has the potential to be utilized and expanded upon future research.
- Minus some limiting assumptions, the authors' identifiability results and inference procedure is valuable.
- The authors provide a rigorous exploration of various aspects of their model in comparison with baselines in synthetic data.

**Weaknesses:**

- The authors make a series of assumptions (1-6) in order to achieve identifiability, with the first one being causal sufficiency. Both the assumption of causal sufficiency and making simplifying assumptions in general is a widely used practice especially when expanding causal analysis to previously unexplored territory, and is usually acceptable to expedite the initial analysis of a new idea. Although it is still understandable that the authors make this assumption, for the kinds of data they are hoping to conduct causal analysis on this assumption unfortunately seems to be especially likely to be violated. For example, it sounds very improbable for measurements made in specific parts of the brain to not have latent confounding. A similar case against disjoint cycles can also be made, while the rest of the assumptions seem less offensive. That being said, I think the authors' contributions justify these limitations.
- Given the presence of these limiting assumptions, some external validation through real data experiments is called for. The authors' experiment with real data present no external validation (e.g. discovering some known causal relationships among various brain regions).

**Questions:**

- In addition to the aforementioned assumptions, using a linear model is another disputable choice, as the authors note. Did the authors experiment with nonlinear generated data to examine the effects of this potential source of model misspecification?
- Brain imaging is an obvious (and worthy) example for the utilization of such a modeling approach. Are there any other impactful application fields that can benefit from such work?
- In L99, is it supposed to be $\mathcal{H}_\ell$ to $\mathcal{H}_j$?
- In L116, is it supposed to be $\\{(t_{ju}, X_{ju})\\}^{m_j}_{u=1}$?

**Limitations:**

The authors generally transparent about their modeling assumptions and potential effects thereof. However some of the points I mentioned above would benefit from more elaboration in the final paper.

---

> ### Author Rebuttal · Authors · 2023-08-10
>
> **Response to your questions**:
>
> **[Q1]** Following your suggestion, we have carried out misspecification analysis for our model by considering nonlinearly generated data. We have considered a scenario with number of samples ($n$) $= 100$, number of nodes ($p$) $= 6$ and evenly spaced time grid over $(0, 1)$ of size $d = 100$. The summary measures corresponding to $10$ replicates are given by:
>
> | Method | TPR | FDR | MCC |
> | :--------: |  :--------: |  :--------: |  :--------: |
> | FENCE | 0.27(0.08) | 0.71(0.07) | 0.34(0.07) |
>
> This poor performance is clearly expected because our modeling assumptions involve linear SEMs. We will add this misspecification analysis result with respect to non linearly generated data in the revised version of the paper.
>
> **[Q2]** Some other application fields that can benefit from our work are (1) finance (e.g., stock price data), (2) mobile health (e.g., wearable device data), and (3) electronic medical records (e.g., longitudinal studies).
>
> **[Q3]** It is a typo. We will correct it in the revision of the paper.
>
> **[Q4]** It is a typo. Instead of $j$ in the subscript, it should be $u$, which runs from 1 to $m\_{j}$. We will correct it in the revision of the paper.
>
> **Response to the weaknesses mentioned**:
>
> 1.(**Assumptions of the model**) We acknowledge the causal sufficiency assumption is limitation of our current work and relaxing it would be an interesting future direction.
>
> For the disjoint cycle assumption, we provide an example below, which shows the reason why the assumption is needed to uniquely identify the underlying graph.
> Consider the following two cyclic graphical models,
>
> **Model 1**:
> \begin{aligned}
>     Y\_{1} &= 0.95Y\_{2} + \epsilon\_{1} \newline
>     Y\_{2} &= Y\_{3} - Y\_{4} + \epsilon\_{2} \newline
>     Y\_{3} &= -1.05 Y\_{1} + \epsilon\_{3}\newline
>     Y\_{4} &= -0.1Y\_{1} + \epsilon\_{4}
> \end{aligned}
>
> **Model 2**:
> \begin{aligned}
>     Y\_{1} &= -0.95Y\_{3} + \epsilon\_{1} \newline
>     Y\_{2} &= 1.05Y\_{1} + \epsilon\_{2} \newline
>     Y\_{3} &= Y\_{2} + Y\_{4} + \epsilon\_{3}\newline
>     Y\_{4} &= -0.1Y\_{1} + \epsilon\_{4}
> \end{aligned}
>
> Both Models 1 and 2 have overlapping cycles. Because independent component analysis (ICA) is identifiable up to the column permutation and scaling of the demixing matrix $(\mathbf{I} - \mathbf{B})^{-1}$, Models 1 and 2 are equivalent as their respective $\mathbf{B}$ matrices are row permutation and scaling of each other.
> Hence, this example shows that in the presence of overlapping cycles, the underlying true graph can only be identified up to (ICA) equivalence classes.
>
> 2.(**Validity of the real data analysis results**) We now discuss the validity of our real data results. In Hayden et al. (2006), it was observed that alcohol-dependent subjects exhibited frontal asymmetry, distinguishing them from the control group. Our own investigation aligns well with these results, as we have identified denser connectivity across various brain regions in the middle and left areas of the frontal lobe among alcoholic subjects, when compared to controls. Furthermore, Winterer et al. (2003) documented coherent differences between alcoholics and controls in the posterior hemispheres, specifically in the temporal, parietal, and occipital lobes. In accordance with their findings, our study provides additional support for this claim, as we have observed heightened activity with several cycles formed in those same regions within the alcoholic group when compared to the control group.
>
> **References**:
>
> * Hayden, Elizabeth P., et al. "Patterns of regional brain activity in alcohol‐dependent subjects." Alcoholism: Clinical and Experimental Research 30.12 (2006): 1986-1991.
>
> * Winterer, G., et al. "EEG phenotype in alcoholism: increased coherence in the depressive subtype." Acta Psychiatrica Scandinavica 108.1 (2003): 51-60.

---

> > ### Comment · Reviewer_nZkc · 2023-08-15
> > **Thanks for the response**
> >
> > I thank the authors for their response and recommend the acceptance of the paper. I think integrating their responses regarding model misspecification, modeling assumptions, and how their results corroborate existing findings into the final version of the paper will further improve their work.

---

> > > ### Author Response · Authors · 2023-08-15
> > >
> > > Thank you for your reply. We are glad that our response helps. We will integrate our responses regarding misspecification analysis, model assumptions and the validation of our real data analysis results in the revised version of the paper.

---

### Official Review · Reviewer_236D · 2023-07-06

**Soundness:** 3 good
**Presentation:** 2 fair
**Contribution:** 2 fair
**Rating:** 5
**Confidence:** 4

**Summary:**

In this paper, the author proposes an operator-based non-recursive linear structural equation based novel causal discovery framework that identifies causal relationships among functional objects in the presence of cycles and additional sampling noises. Furthermore, author demonstrates the effectiveness of proposed method from experiment and theory.

**Strengths:**

The motivation is clear and the writing is well. The experiment results show that the proposed method significantly outperforms the baselines. The causal identifiability proof is impressive.

**Weaknesses:**

Please refer to the Questions.

**Questions:**

1. The proposed year of your baselines fLiNG, LiNGAM, PC, CCD are 2022, 2006, 1991, 1996 respectively, more recent baselines should be considered. According to line 282 and 283, the codes of Lee (2022) and Yang (2022) are not available. However,  fLiNG’s code is also not public (I tried my best to find it but failed). Hence, how do you obtain the results reported in the paper ?
2. In line 18, you said casual discovery is popular in machine learning. But both in introduction and related work, I do not find relative description to support this standpoint.
3. How to obtain equation (3) ? It appears to be left-multiplying Equation (1) by  and  respectively. If so, the term in first row should be .
4. In line 116, Why does j vary from 1 to  ? The range of j should be from 1 to p.
5. Many symbols are used without definition, like Beta in line 241, Dir and IG in line 254. Some symbols are easy to search for their meanings, while others are not as straightforward.
6. The equation (7) are not mentioned in Section 3, is it really necessary ?
7. The name of Section 3 is Model inference. However, nothing but the prior distribution of parameters are introduced, this is confusing.
8. Reference [Fangting Zhou, Kejun He, and Yang Ni. Causal discovery with heterogeneous observational data] mentions that they do not restrict their model to be acyclic, which is in conflict with your statement in line 56 (Related work section).

**Limitations:**

No apparent negative societal impacts.

---

> ### Author Rebuttal · Authors · 2023-08-09
>
> **Response to your questions**:
>
> **[Q1]** We obtained the fLiNG's code from the authors of that paper. Our research team has also reached out to Yang (2022) but we were unable to obtain their code. The paper by Lee (2022) mainly focused on the statistical theories and hence the code is not available.
>
> **[Q2]** Causal discovery has seen applications in many machine learning areas such as causal representation learning, causal fairness, causal transfer learning, and causal reinforcement learning.
>
> * Bernhard Schölkopf, Francesco Locatello, Stefan Bauer, Nan Rosemary Ke, Nal Kalchbrenner, Anirudh Goyal, Yoshua Bengio "Toward causal representation learning." Proceedings of the IEEE 109.5 (2021): 612-634.
>
> * Tang, Zeyu, Jiji Zhang, and Kun Zhang. "What-Is and How-To for Fairness in Machine Learning: A Survey, Reflection, and Perspective." ACM Computing Surveys 55.13s (2023): 1-37.
>
> * Mateo Rojas-Carulla, Bernhard Schölkopf, Richard Turner, Jonas Peters "Invariant models for causal transfer learning." The Journal of Machine Learning Research 19.1 (2018): 1309-1342.
>
> * Yan Zeng, Ruichu Cai, Fuchun Sun, Libo Huang, Zhifeng Hao. "A Survey on Causal Reinforcement Learning." 	arXiv:2302.05209 (2023).
>
> We will add these references in the revision of our paper.
>
>
> **[Q3]**
> \begin{aligned}
> \mathcal{P}\_{j}Y\_{j} &= \sum\_{\ell}\mathcal{P}\_{j}\mathcal{B}\_{j\ell}Y\_{\ell}~ + \mathcal{P}\_{j}f\_{j} \newline
> &= \sum\_{\ell}\mathcal{P}^{2}\_{j}\mathcal{B}\_{j\ell}\mathcal{P}\_{\ell}Y\_{\ell}~ + \mathcal{P}\_{j}f\_{j} \newline
> &= \sum\_{\ell}\mathcal{P}\_{j}\mathcal{B}\_{j\ell}\mathcal{P}^{2}\_{\ell}Y\_{\ell}~ + \mathcal{P}\_{j}f\_{j} \newline
> &= \sum\_{\ell}\mathcal{B}\_{j\ell}\mathcal{P}\_{l}Y\_{\ell}~ + \mathcal{P}\_{j}f\_{j}
> \end{aligned}
>     The second equality is obtained since we assume ${\mathcal{B}}\_{j\ell}=\mathcal{P}\_{j}\mathcal{B}\_{j\ell}\mathcal{P}\_\ell$ (stated above equation (3)), which means that causal effects can be fully described within the low-dimensional subspace $\mathcal{D}\_{j}$. The third equality follows from that fact that $\mathcal{P}\_{j}$ is a projection onto $\mathcal{D}\_{j}$ and therefore we have $\mathcal{P}^{2}\_{j} = \mathcal{P}\_{j} \forall j\in$\{$1, \cdots, p$\}. The fourth equality is obtained again by using ${\mathcal{B}}\_{j\ell}=\mathcal{P}\_{j}\mathcal{B}\_{j\ell}\mathcal{P}\_\ell$.
>
> **[Q4]** It is a typo. Instead of $j$ in the subscript, it should be $u$, which runs from 1 to $m\_{j}$. We will fix the typo in the revision of the paper.
>
> **[Q5]** By Beta, Dir and IG, we meant beta distribution, Dirichlet distribution, and inverse-gamma distribution, respectively. We will carefully read our paper and fix all the symbols and short-hands in the revision of the paper.
>
> **[Q6]** We referred to equation (7) in the proof for causal identifiability in Section A of the supplementary material. We also used the term $\mathbf{\Phi}(\mathbf{t})$ defined in equation (7) for the description of Assumption 6 in the paper.
>
> **[Q7]** We agree and we will change the title of Section 3 from "Model Inference" to "Bayesian Model Formulation".
>
> **[Q8]** We believe the references you mentioned are from lines 54 and 56 in the paper. Since the first author has two papers in 2022, namely 2022a and 2022b, we made a reference to their paper titled 'Functional Bayesian Networks for Discovering Causality from Multivariate Functional Data' (2022b) in line 54. In this particular paper, the author constrains their model to be acyclic. Note that in their paper (2022a), which we cited in a different place, they did consider cyclic graphs.

---

> > ### Comment · Reviewer_236D · 2023-08-11
> > **Comments after Rebuttal**
> >
> > Thanks for addressing my issues, Although there are still certain defect to be improved in this paper, the motivation is very interesting, and the proposed behaviour is technical solid. Concretely, I prefer to change my rating.

---

> > > ### Author Response · Authors · 2023-08-13
> > >
> > > Thank you for your reply. We are glad that our response helps. We will revise our work accordingly.

---

### Official Review · Reviewer_ucHi · 2023-07-12

**Soundness:** 4 excellent
**Presentation:** 4 excellent
**Contribution:** 3 good
**Rating:** 7
**Confidence:** 4

**Summary:**

The paper considers the problem of learning causal structure from multivariate functional data that may involve cyclic interactions.
It employs an adaptive mapping to a lower dimensional space that retains the relevant causal information, in combination with a Bayesian framework that obtains posterior estimates through MCMC sampling. Effectiveness of the model is compared against several alternatives on synthetic and real-world data.

**Strengths:**

The paper is exceptionally well written and contains a comprehensive and powerful approach to cyclic causal discovery.

Many of the steps involved are not new by themselves, but brought together effectively to produce an elegant and novel method that shows promising performance. The descriptions and theoretical derivations are concise, clear and consistent, and each of the steps involved follows logically to reduce a challenging problem to an effective, fully Bayesian solution. I particularly liked the way the adaptive basis expansion in 3.2 was incorporated into the overall method.

Assumptions are strong, but clearly and carefully spelled out, leading to the desired identifiability conclusion in Thm2.1.

The synthetic data experimental evaluation is convincing (although due to the required assumptions inevitably biased in favour of the proposed approach), and the alcoholic EEG application is interesting (though hard to evaluate for non-experts).

**Weaknesses:**

- although I really like the paper, I think the main contribution lies mostly in the way the various steps are brought together in a clear and coherent way than in fundamentally new insights or approaches. (Nevertheless, I do consider the end result a valuable and novel contribution.)
- assumptions are quite strong: to (still) see causal sufficiency feature so prominently is disappointing, the ‘disjoint cycles’ should be unnecessary (as admitted by the authors), and ‘non-Gaussianity’ may seem generic but implies identifiability may rely on weak distributional signals (data hungry) and it excludes the challenging complications encountered in the linear Gaussian case that e.g. CCD was specifically designed to handle.

- one confusing aspect (at first) was that in the beginning of the paper the introduction of the time measurements seemed to suggest we are doing time-series analysis, but on closer inspection it is actually closer to standard observational causal inference (with cycles), right?

- some questionable claims, e.g. l.245 ‘strongly prevents false discoveries’ could equally be stated as ‘heavily biased towards sparsity’, and l.307 ‘strong evidence of the effectiveness of FENCE compared to existing methods’ should contain the caveat ‘under the stated assumptions 1-6’. In particular, starting from PCA for LINGAM / CCD seems questionable, where LiNGAM is designed for DAGs, and independence based methods like CCD do not (need to) rely on functional assumptions that are essential for FENCE.


**Questions:**

- what is the relation/difference between your model assumptions and e.g. the simple SCMs in Bongers et al. [‘Foundations of structural causal models with cycles and latent variables.’,2018]?

- how do you determine Kj for the lower dimensional space embedding?

- is it true that you essentially treat multiple observations at different time points as independent observations on different system instances in equilibrium?

- in. Table 1, how do you compare non-invariant edges in the equivalence class of e.g. PC/CCD vs. the ground truth? (as ‘half wrong’?)
- given the causal sufficiency assumption, how come you find bidirected edges in Fig.2?

Other remarks (for lack of a better place to put them):
- title seems grammatically a bit weird
- 5 ‘enhance interpretability’ => I understand the practical effectiveness in going via a lower dimensional internal representation, but the output does not reflect this, right? So how does it affect ‘interpretability’?
- 88 ‘cyclic component’ => this definition seems a bit off as it allows for vertices in a subgraph that are not part of a cycle (so only the full graph G can be a ‘maximal’ cyclic component). I would suggest using the standard term ‘strongly connected component’.
- 258 ‘simulate posterior samples through MCMC’ => actually I would not mind seeing a bit more on this step in the main paper

---

> ### Author Rebuttal · Authors · 2023-08-10
>
> **Response to your questions**:
>
> **[Q1]** Bongers introduces simple structural causal models (SCMs), which allows certain cyclic settings. An SCM is simple if any subset of its structural equations can be solved uniquely for its associated variables in terms of the other variables that appear in these equations.
>
> * **Similarity in assumptions**:  Like our Assumption 2, simple SCMs do not allow for self cycles.
>
> * **Dissimilarity in assumptions**:
>     1. As we focus on linear SEMs, Proposition C.3 in Appendix C of their paper states that a necessary and sufficient condition for a linear SCM to have unique solvability w.r.t. a subset $\mathcal{L} (\subseteq V)$ is the invertibility of the matrix $\mathbf{A}\_{\mathcal{L}\mathcal{L}} = \mathbf{I} - \mathcal{B}\_{\mathcal{L}\mathcal{L}}$. A following remark also states that a sufficient condition for invertibility of $\mathbf{A}\_{\mathcal{L}\mathcal{L}}$ is that the spectral radius of the matrix $\mathbf{B}\_{\mathcal{L}\mathcal{L}}$ is less than $1$. So for a linear SCM to be simple, this condition needs to be true for all such subsets $\mathcal{L}$. Although this condition implies the stability assumption (Assumption 3) of our paper, the converse is not true. So our stability assumption is less stringent.
>     2. They allowed confounders  whereas we assume causal sufficiency (Assumption 1).
>     3. One key property of simple SCMs is that the solutions always satisfy the conditional independencies implied by $\sigma$-separation. So most nonparametric independence test-based methods can be used for simple SCMs by replacing $d$-separation with $\sigma$-separation. This allows the graph to be identified up to an equivalence class. But, in our paper, additional parametric assumptions (Assumption 4 \& 5) allows us to identify a unique graph.
>
>
> **[Q2]**
> To determine $K_j$, we adopted a simple heuristic approach, following Kowal et al. (2017) and Zhou et al. (2022), which selects conservative global value for $K\_{j} = K$ for all $j \in {1, \dots, p}$. Specifically, we impute the functional observations and arrange them into a $(n \times p) \times d$ matrix, where $d = |\cup\_{i,j} \mathcal{T}^{(i)}\_{j}|$ is the size of the union of the measurement grid over all functions. Next, we perform SVD and select the minimum value of $K$ to explain at least 90\% of the variance. Note that although $K$ is fixed throughout MCMC, the basis functions are inferred (i.e., they are not functional principal components) adaptively to causal structure.
>
> Now, we've implemented another approach to tune $K$ based on WAIC. We tested it in simulations. We found that the $K$ value resulting in the lowest WAIC varies mostly around $\pm2$ from the value we obtained using the heuristic approach. The graph recovery performance measured by Matthew's correlation coefficient is pretty similar to that of the heuristic approach. We will discuss these in the revised paper.
>
> **[Q3]** We assume our data are drawn from an equilibrium distribution of a dynamic system involving multivariate functions.
> Let $\mathbf{Y}(\cdot)[t]=(Y\_1(\cdot)[t],\dots,Y\_p(\cdot)[t])^\top$ denote a vector of functions at time point $t$ where the domain of each function $Y_j(\cdot)[t]$ is not necessarily time. Note that we have used $()$ to denote the function input/domain and $[]$ to denote the time index of a dynamic system.
> We consider an AR1-type dynamic system of those functions,
>
> $$\mathbf{Y}(\cdot)[t] = \mathbf{\mathfrak{B}}\mathbf{Y}(\cdot)[t-1] + \mathbf{f}(\cdot)$$
>
> where $\mathfrak{B} = (\mathcal{B}\_{j\ell})\_{j,\ell = 1}^{p}$ is a matrix of operators and $\mathbf{f}(\cdot)$ is a $p$-variate noise process, which has the same domain as $\mathbf{Y}(\cdot)[t]$. This recursively leads to,
> $$\mathbf{Y}(\cdot)[t] = \mathbf{\mathfrak{B}}^{t}\mathbf{Y}(\cdot)[0] + \sum\_{s = 0}^{t-1} \mathbf{\mathfrak{B}}^{s}\mathbf{f}(\cdot)$$
> For the system to eventually reach equilibrium, it is crucial for both $\mathbf{\mathfrak{B}}^{t}$ and $\sum\_{s=0}^{t-1}\mathbf{\mathfrak{B}}^{s}$ to converge as $t$ approaches infinity. The condition essential for this convergence is that the moduli of the eigenvalues of $\mathbf{\mathfrak{B}}$ must be less than 1, and none of the real eigenvalues should equal 1. Given this criterion, as $t \rightarrow \infty$, $\mathbf{\mathfrak{B}}^{t}$ approaches 0, and $\sum\_{s=0}^{t-1}\mathbf{\mathfrak{B}}^{s}$ tends towards $(\mathbf{I} - \mathbf{\mathfrak{B}})^{-1}$. Consequently, we can deduce that as $t\to \infty$,
> $$\mathbf{Y}(\cdot)[t] \rightarrow \mathbf{Y}(\cdot) = (\mathbf{I} - \mathbf{\mathfrak{B}})^{-1}\mathbf{f}(\cdot)$$ which is equivalent to Equation (2) in our paper. Therefore, we model $\mathbf{Y}(\cdot)$ at the equilibrium of a functional dynamic system.
>
> **[Q4]** We favored PC and CCD by counting a non-invariant edge between two nodes as a true positive as long as the two nodes are adjacent in the true graph.
>
> **[Q5]** The bi-directed edges shown in Figure 2 are directed cycles, i.e., $i\leftrightarrow j$ means $i\to j$ and $j\to i$. We will explain it in the revised paper.
>
> **Response to the weaknesses mentioned**:
>
> * (**Confusion regarding time measurements**) Yes, you are right. By viewing an entire function as one random object, we essentially discover causality based on cross-sectional observational data even though the functional object may be a function of time. In fact, they need not be functions of times; they could have other domains such as space and frequency domains.
>
> * (**Some questionable claims**) Yes, we agree to your point regarding the false discovery; we will change it to "prevents false discoveries and leads to a sparse network" in the revised paper.
>     We also agree with you that LiNGAM, PC, and CCD are not the best methods for the cyclic functional causal discovery problem. However, we had to resort to these methods for comparison because of the general lack of existing methods for this problem.

---

### Author Rebuttal · Authors · 2023-08-10

We would like to thank all reviewers for their constructive comments.
This is a global response to several points mentioned by different reviewers while reviewing our work.

1. **Assumptions of the model**: Some of the reviewers pointed out limitations regarding some of the assumptions that we have taken to show causal identifiability of our model in the paper. We acknowledge that the causal sufficiency assumption is limitation of our current work and relaxing it would be an interesting future direction. As for the disjoint cycle assumption, we provide an example below, which indicates that allowing overlapping cycles could result in non-identifiability.
Consider the following two cyclic graphical models,

**Model 1**:
\begin{aligned}
    Y\_{1} &= 0.95Y\_{2} + \epsilon\_{1} \newline
    Y\_{2} &= Y\_{3} - Y\_{4} + \epsilon\_{2} \newline
    Y\_{3} &= -1.05 Y\_{1} + \epsilon\_{3}\newline
    Y\_{4} &= -0.1Y\_{1} + \epsilon\_{4}
\end{aligned}

**Model 2**:
\begin{aligned}
    Y\_{1} &= -0.95Y\_{3} + \epsilon\_{1} \newline
    Y\_{2} &= 1.05Y\_{1} + \epsilon\_{2} \newline
    Y\_{3} &= Y\_{2} + Y\_{4} + \epsilon\_{3}\newline
    Y\_{4} &= -0.1Y\_{1} + \epsilon\_{4}
\end{aligned}

Both Models 1 and 2 have overlapping cycles. Because independent component analysis (ICA) is identifiable up to the column permutation and scaling of the demixing matrix $(\mathbf{I} - \mathbf{B})^{-1}$, Models 1 and 2 are equivalent as their respective $\mathbf{B}$ matrices are row permutation and scaling of each other.


2. **Validation of our real data analysis results**: Some of the reviewers wanted to understand the validation of our real data analysis results. So we have found out references from existing literature (Hayden et al. (2006), Winterer et al. (2003)) that supports our claims in the real data analysis results and thus have provided validations of our results. We will add those references in the revised version of the paper.

3. **Comparison with the baselines for real data analysis**: We have attached a pdf showing the brain connectivity of alcoholic and control groups for the baseline methods fPCA-PC and fPCA-CCD. We could not carry out fPCA-LiNGAM because LiNGAM is not applicable to cases where $q > n$ with $q = Kp$ being the total number of extracted basis coefficients across all functions, a problem that we mentioned in the simulation studies as well. The brain connectivity for fLiNG is shown in the paper Zhou et al(2022). While we could not provide a clear image close to what we have in the paper due to time constraints, some of the key points that we can gather from these brain connectivity plots are:
    1. fPCA-PC could find out denser connectivity across the temporal regions of the brain in the alcoholic group compared to that of the control group.
    2. Apart from finding denser connectivity in the left temporal lobe of brain in the alcoholic group compared to the control group, fPCA-CCD could find some of the two way cycles ($X \rightarrow Y$ and $Y \rightarrow X$) for the nodes that are adjacent to each other.

4. **Typos**: We have acknowledged the typos that the reviewers have pointed out to us and we would make those corrections in the revised version of the paper.

**Reference**:

1. Zhou, Fangting, et al. "Functional Bayesian Networks for Discovering Causality from Multivariate Functional Data." arXiv preprint arXiv:2210.12832 (2022).

2. Hayden, Elizabeth P., et al. "Patterns of regional brain activity in alcohol‐dependent subjects." Alcoholism: Clinical and Experimental Research 30.12 (2006): 1986-1991.

3. Winterer, G., et al. "EEG phenotype in alcoholism: increased coherence in the depressive subtype." Acta Psychiatrica Scandinavica 108.1 (2003): 51-60.

---

### Comment · Reviewer_ucHi · 2023-08-18

The other reviews and author rebuttals have strengthened my original impression that this is an interesting and worthwhile contribution that deserves to be in the conference. Hence I will stay with 'Accept'.

---

### Decision · Program_Chairs · 2023-09-21

**Decision:**

Accept (poster)

**Comment:**

The reviewers appreciated both the quality of the paper and the results, aimed at learning causal models involving cycles.

Under the usual limiting assumption of causal sufficiency, and further assuming that the causal cycles are disjoint, the use of the projection on a lower dimensional domain is shown to open a new avenue to deal with causal cycles. The dialog with the reviewers convincingly clarified the merits of the approach compared to the state of the art (NoTEARS and DAG-GNN), discussed the hyper-parameter adjustment, and discussed the relevance of the results on EEG data (though these may violate the second assumption).